# PortraitRL: Reinforcement Learning for Portrait Pose Transfer with Multi-Objective Reward

Jiahui Wu [*1]   Zelong Sun [*1]   Yanbiao Ma [1]   Zhiwu Lu [1]

## Abstract

Portrait pose transfer (PPT) requires generative models to preserve fine-grained identity details while following complex pose and layout modification instructions. Existing methods often struggle with extensive data annotation requirements or employ optimization objectives that are suboptimal for addressing PPT's two key challenges. In this work, we propose **PortraitRL**, a novel post-training framework that addresses these challenges with a multi-objective reward mechanism. Specifically, we employ LVLM-based reward functions to effectively evaluate PPT's two challenges and apply within-group standardization to eliminate scale differences, allowing these rewards to effectively guide optimization. More importantly, we devise a novel reinforcement learning algorithm, Negative-aware Score Preference Optimization (**NaSPO**), which automatically identifies positive and negative preference samples through within-group advantages, eliminating annotation requirements while fully leveraging both positive and negative learning signals. Extensive experiments show state-of-the-art performance, with significant improvements in both detail preservation and editing accuracy.

## 1. Introduction

In the multimedia era, high-quality portrait photographs have become an essential medium for personal expression and branding. This has given rise to a novel and challenging image editing task: Portrait Pose Transfer (PPT) (Sun et al., 2026; Peng et al., 2024; Ye et al., 2023). As illustrated in Figure 1, conventional image editing typically involves simple modifications with fixed layouts. In contrast, PPT presents two unique challenges: (1) **Complex modification instruction following**. Generated images should follow complex

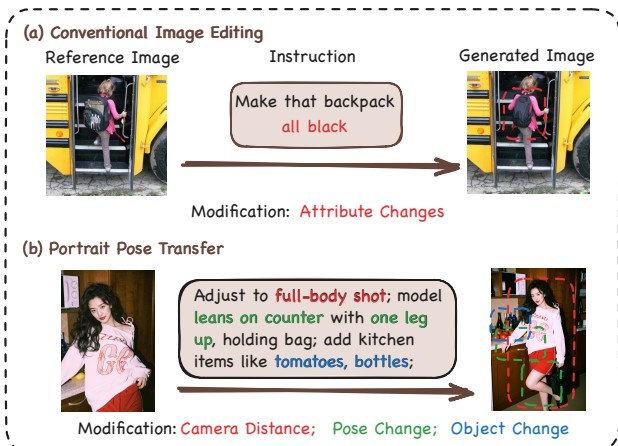

*Figure 1.* **Conventional image editing vs. Portrait Pose Transfer (PPT).** Unlike conventional editing with simple instructions and fixed layouts, PPT handles complex pose and layout modifications while preserving fine-grained identity details.

pose and layout modification instructions (Zhang et al., 2023; Li et al., 2024a). (2) **High-fidelity detail preservation.** Generated images should faithfully retain fine-grained details from the reference image, such as makeup, clothing, and other visual characteristics (Ye et al., 2023).

Current methods (Ruiz et al., 2023; Zhang et al., 2023) primarily rely on Supervised Fine-Tuning (SFT) to learn this transformation using vast amounts of annotated triple data (reference image, editing instruction, and target image). However, acquiring high-quality image pairs with consistent identities at scale is often prohibitively difficult in practice. Moreover, PPT is intrinsically a one-to-many mapping problem, where a single pose instruction can correspond to multiple valid spatial layouts. The rigid single-target supervision in SFT over-penalizes reasonable structural variations that deviate from the specific ground truth, thereby constraining both model generalization and output diversity.

Recently, Reinforcement Learning (RL) (Shao et al., 2024; Rafailov et al., 2023; Ethayarajh et al., 2024; Dai et al., 2026) has been introduced to the image generation domain (Xu et al., 2023; Wu et al., 2023) to align model behaviors with complex human preferences and improve the perceptual quality of generated content. DPO-based approaches (Wallace et al., 2024; Zhu et al., 2025) leverage

---
[*]Equal contribution [1]Gaoling School of Artificial Intelligence, Renmin University of China, Beijing, China. Correspondence to: Zhiwu Lu <luzhiwu@ruc.edu.cn>.

*Proceedings of the 43rd International Conference on Machine Learning*, Seoul, South Korea. PMLR 306, 2026. Copyright 2026 by the author(s).

annotated preference data to guide optimization, yet they still face the challenge of data annotation difficulty. GRPO-based methods (Liu et al., 2025a; Xue et al., 2025) reduce annotation requirements by comparing samples within the same generation group. However, these methods' reward functions are primarily designed for text-to-image generation tasks and lack effective mechanisms for complex editing scenarios, such as PPT. Furthermore, most approaches employ CLIP-based models (Radford et al., 2021; Lu et al., 2022; Wu et al., 2023; Jing et al., 2024) as reward models, which can only capture coarse-grained semantic alignment between the generated image and condition inputs. However, PPT demands highly accurate instruction following and fine-grained detail preservation, rendering such global similarity metrics inadequate.

To address these challenges, we propose **PortraitRL**, a novel post-training framework for PPT with a Multi-Objective Reward Mechanism. Specifically, to tackle PPT's two unique challenges of complex instruction following and fine-grained detail preservation, we design two dedicated reward objectives: **prompt following** and **detail preservation**. Unlike conventional CLIP-based rewards that only capture global similarity, we employ Large Vision-Language Model (LVLM) based score functions that enable fine-grained evaluation. These LVLM-based scores simulate human judgment criteria, providing human-preference assessments for generated images. We further apply within-group standardization to eliminate scale differences across both objectives. We also devise **N**egative-**a**ware **S**core **P**reference **O**ptimization (**NaSPO**), a novel GRPO-based RL algorithm optimized with both positive and negative preference scores. Unlike DSPO (Zhu et al., 2025), our NaSPO identifies positive and negative preferred samples through within-group relative advantages, eliminating the need for manual annotation. Besides the positive preference score, NaSPO additionally defines a negative preference score by leveraging information from negative preferred samples, creating a dual-preference mechanism. Extensive experiments demonstrate that PortraitRL achieves superior performance compared to existing baselines, with significant improvements in both detail preservation and pose accuracy.

Our main contributions can be summarized as follows: **(1)** We propose PortraitRL, a GRPO-based post-training framework tailored for the PPT task. **(2)** We introduce a Multi-Objective Reward Mechanism which consists of two reward tasks with LVLM-based score functions and within-group standardization. **(3)** We devise NaSPO, a novel RL algorithm that leverages both positive and negative preference scores to guide optimization, eliminating the need for annotated preference data. **(4)** Extensive experiments show that PortraitRL achieves significant improvements in both detail preservation and editing instruction following.

## 2. Related Work

### 2.1. Portrait Pose Transfer

Since the advent of Denoising Diffusion Probabilistic Models (DDPM) (Ho et al., 2020; Yang et al., 2024), the field of image generation has witnessed rapid progress. However, the escalating demand for controllable generation has given rise to novel challenges, such as portrait pose transfer, which require maintaining high fidelity to both visual prompts and textual instructions. To address the challenges, pioneering approaches have been proposed: DreamBooth (Ruiz et al., 2023) utilizes unique identifier tokens; ControlNet (Zhang et al., 2023) introduces supplementary spatial control branches; and IP-Adapter (Ye et al., 2023) employs decoupled cross-attention mechanisms. These methods effectively inject identity and structural features while preserving the text-to-image generation capabilities of the base model. More recently, the emergence of Flow Matching and Rectified Flow (Lipman et al., 2022) paradigms, which utilize vector fields as the primary training objective, has shown superior performance compared to previous iterations. This theoretical advancement has culminated in state-of-the-art architectures such as FLUX.1-Kontext [Dev] (Batifol et al., 2025) and Qwen-Image-Edit (Wu et al., 2025). Built upon the flow-matching framework, these models leverage dual text-image inputs to achieve exceptional context awareness and fine-grained identity preservation in complex editing scenarios.

### 2.2. Reinforcement Learning

Effectively integrating reinforcement learning (RL) into diffusion models is a promising yet challenging research direction. Conventional text-to-image diffusion models are usually trained in a single stage using large-scale text–image pairs with noise-prediction objectives. Although such models achieve high visual quality, their alignment with user intent or task-specific requirements remains limited. Inspired by the "pre-training + RLHF" paradigm in large language models, recent studies have begun to introduce RL-based fine-tuning into diffusion models to enhance preference alignment. To align with human preferences, existing works have curated preference datasets and developed relevant reward models, such as ImageReward (Xu et al., 2023), PickScore (Kirstain et al., 2023), HPSv2 (Wu et al., 2023), and CLIP (Radford et al., 2021), to improve semantic alignment, aesthetic quality, and instruction following.

Despite this progress, applying RL to diffusion models still faces major challenges. Reward modeling is costly and difficult, as it requires large-scale, high-quality annotations and must capture subjective and complex visual preferences. Moreover, most RL strategies are adapted from LLMs, while diffusion models differ fundamentally in generation dynamics and training objectives, making direct transfer subopti-

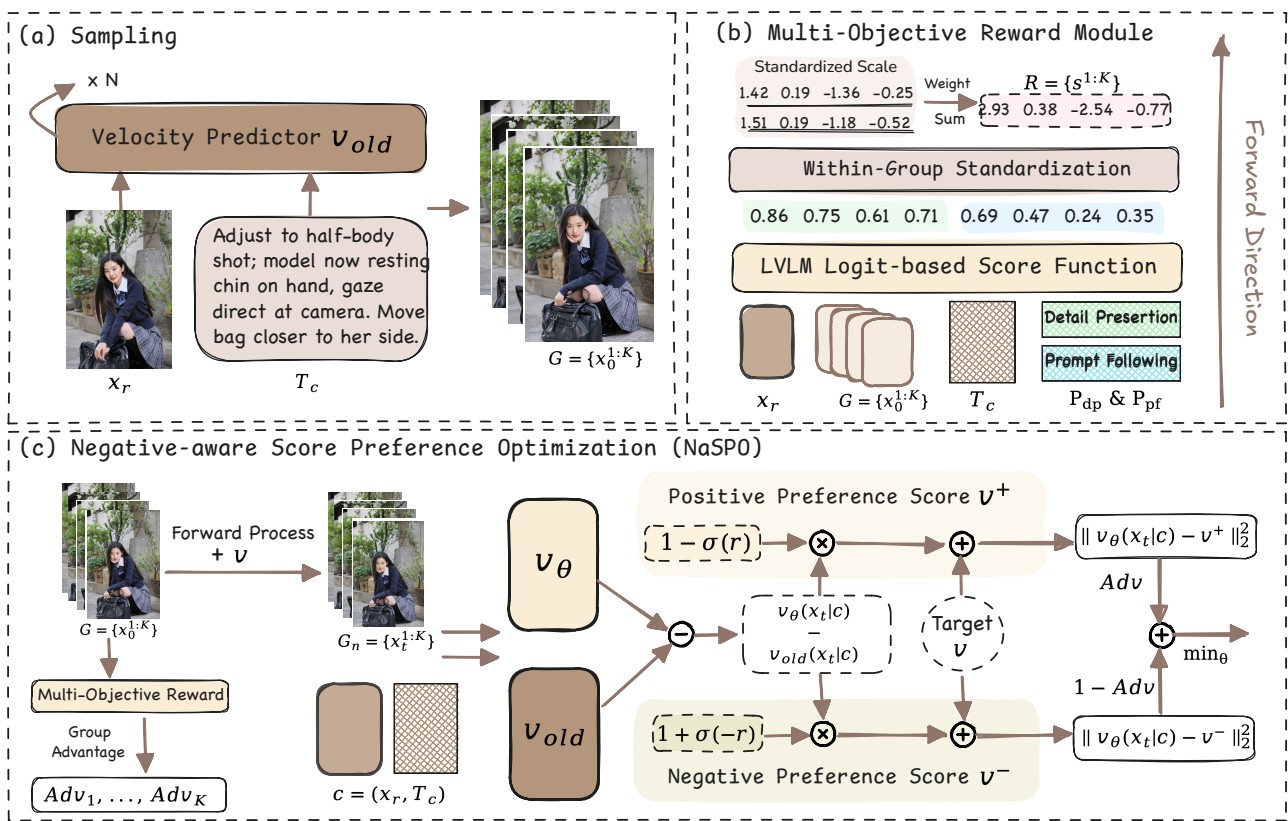

*Figure 2.* **Overview of PortraitRL pipeline.** (a) Given a reference image and editing instruction, we first sample a group of generated images. (b) We then calculate the reward through a Multi-Objective Reward Mechanism. (c) Finally, we design NaSPO RL frame work with both positive and negative preference score to optimize the model.

mal. To address these issues, a series of alignment methods have been proposed, including ReFL (Xu et al., 2023) for reward feedback learning, Diffusion-DPO (Wallace et al., 2024) and Diffusion-KTO (Li et al., 2024b) based on preference optimization, DSPO (Zhu et al., 2025) via score function, GRPO-based methods (Liu et al., 2025a; Xue et al., 2025) within flow-matching frameworks, and Diffusion-NFT (Zheng et al., 2025) with contrastive optimization. Although these methods demonstrate the feasibility of RL-based alignment for diffusion models, there is still substantial room for improvement, especially for high-precision tasks such as image editing.

## 3. Methodology

### 3.1. Preliminaries

**Portrait Pose Transfer.** Given a reference image $x_r$ and a textual pose transfer instruction $T_c$, the objective of PPT is to generate an image $x_0$ that simultaneously preserve key attributes from $x_r$ such as identity and appearance and follow the pose transfer described in $T_c$.

**Flow Matching.** Flow Matching is based on the concept of vector fields. Given a data sample $x_0 \sim X_0$ from a target

distribution, a noise sample $x_1 \sim X_1$ from a Gaussian distribution, and a condition $c$, the intermediate state $x_t$ is defined as (Liu et al., 2022; Lipman et al., 2022):

$$x_t = (1-t)x_0 + tx_1. \tag{1}$$

For any time step $t$, the training objective is to optimize the network $v_\theta(x_t, t, c)$ to approximate the vector field $v = x_1 - x_0$. The loss function is given by:

$$\mathcal{L}(\theta) = \mathbb{E}_{t, x_0 \sim X_0, x_1 \sim X_1}\left[\|v - v_\theta(x_t, t, c)\|^2\right]. \tag{2}$$

**DSPO in Flow Matching.** DSPO (Zhu et al., 2025) aligns diffusion models with human preferences by decomposing the conditional score function into a pretrained prior term and a preference guidance term:

$$\nabla_{x_t} \log p_\theta(x_t \mid c, y) = \nabla_{x_t} \log p(x_t \mid c) \\ + \gamma \nabla_{x_t} \log p(y \mid x_t, c), \tag{3}$$

where $c$ is the condition, $y$ represents the preference signal, and $\gamma$ is a guidance scale. The preference probability is modeled as $p(y \mid x_t, c) = \sigma(r(c, x_t) - r(c, x_t^l))$, where $x_t^l$ denotes a negative sample and $r(\cdot, \cdot)$ is the reward model. To avoid training an extra reward model, DSPO derives an

implicit reward $r(\cdot, \cdot)$ based on the discrepancy between the current and reference models:

$$r(x_t, c) = \frac{-\lambda\beta_{t+1}}{2(1-\bar{\alpha}_t)} \frac{\alpha_t}{\alpha_{t+1}} \left( \left\| \epsilon_\theta(x_{t+1}, t+1) - \epsilon_{t+1} \right\|_2^2 \right.$$
$$\left. - \left\| \epsilon_{\text{ref}}(x_{t+1}, t+1) - \epsilon_{t+1} \right\|_2^2 \right). \quad (4)$$

Substituting this implicit reward into the objective, the final loss function is defined as:

$$\mathcal{L}_{\text{DSPO}}^t = A(t) \left\| \epsilon_{\theta,t+1} - \epsilon_{t+1} - \lambda\gamma \left( 1 - \sigma\big(r(c, x_t) - \right.\right.$$
$$\left.\left. r(c, x_t^l)\big)\right) \cdot (\epsilon_{\theta,t+1} - \epsilon_{\text{ref},t+1}) \right\|_2^2, \quad (5)$$

where $A(t)$ is a weighting function and $\epsilon_{\text{ref}}$ denotes the noise prediction from the reference model.

While Equation (5) formulates the loss for diffusion models, we apply an analogous derivation to the Flow Matching framework. Instead of presenting the minimization objective, we directly provide the optimal solution as follows (see Appendix C for the detailed derivation):

$$v_{\theta,t+1} = v_{t+1} + \big(1 - \sigma(r(c, x_t)$$
$$- r(c, x_t^l))\big)(v_{\theta,t+1} - v_{\text{ref},t+1}). \quad (6)$$

Leveraging the equivalence between noise vector and velocity field established in Flow-DPO (Liu et al., 2025b):

$$\|\epsilon_{t+1} - \epsilon_{\theta,t+1}\|^2 = (1-t)^2 \|v_{t+1} - v_{\theta,t+1}\|^2, \quad (7)$$

we obtain the final $r(c, x_t)$ in flow matching:

$$r(c, x_t) = -\frac{\beta_t}{2}(\|v_{t+1} - v_{\theta,t+1}\|^2 - \|v_{t+1} - v_{\text{ref},t+1}\|^2). \quad (8)$$

## 3.2. Multi-Objective Reward Mechanism

Reward functions aim to effectively evaluate generated results and provide supervision signals for model optimization. Since PPT requires simultaneously assessing visual detail consistency between generated results and reference images, as well as completion of complex modification instructions, conventional CLIP-based reward functions that evaluate only from global similarity making them unsuitable for PPT. Inspired by (Li et al., 2025), we design an LVLM logit-based reward function that leverages the fine-grained understanding of LVLMs for more precise evaluation.

**LVLM Logit-based Reward Function.** As illustrated in Figure 2, given the input $X$ to be evaluated and an evaluation prompt $P$, the LVLM model generates a score for $X$ guided by $P$. To enable fine-grained evaluation, we partition the score space into $n$ discrete levels $M = \{m_0, \dots, m_n\}$. However, directly using the discrete score would yield

sparse reward signals, failing to capture the model's uncertainty in its assessment. To address this limitation, we compute scores through logits, which provide continuous and smooth reward signals. Specifically, we first obtain the predicted probability for each score level $m_i \in M$ via LVLM:

$$\{L_{m0}, \dots, L_{m_i}\} = \text{LVLM}(X, P), \quad (9)$$

where $L_{m_i}$ denotes the logit of score $m_i$ extracted from the LVLM's generated token. We then compute the expected score as the final evaluation:

$$S = \sum_{i=0}^{|M|} m_i \cdot L_{m_i}. \quad (10)$$

This score captures the model's confidence distribution across scores. Then we normalize scores to the range $[0, 1]$:

$$s = \frac{S - \min(M)}{\max(M) - \min(M)}, \quad (11)$$

where $\min(\cdot)$ and $\max(\cdot)$ return the minimum and maximum values in the set, respectively.

**Multi-Objective Reward.** As discussed above, PPT faces two key challenges: visual detail preservation and complex instruction following. To address these challenges, as illustrated in Figure 2, we design two evaluation prompts, $P_{dp}$ and $P_{pf}$, which guide the model to evaluate these two critical dimensions, respectively. Specifically, for a generated image $I_g^i$, we construct the evaluation input $X_{dp} = (x_r, x_0^i)$ and evaluation prompt $P_{dp}$, and compute the LVLM logit-based reward $s_{dp}^i$ to assess the visual detail preservation of the generated image. Similarly, through $X_{pf} = (x_r, T_c, x_0^i)$ and $P_{pf}$, we can obtain $s_{pf}^i$ to evaluate the complex instruction following capability.

**Within-Group Standardization.** However, due to the inherent difficulty disparity between the evaluation tasks, these two reward scores may exhibit significantly different distributions. When directly aggregating these scores through simple addition, this distribution mismatch may leads to a "reward hacking" phenomenon, where the model preferentially optimizes for the easier metric while neglecting the more challenging one, since the easier metric scores dominate the combined reward signal.

To address this issue and ensure commensurability between the two dimensions, we apply within-group standardization using Z-score normalization. Specifically, as illustrated in Figure 2, for a given reference image and modification instruction pair $(x_r, T_c)$, we first sample a group of generated images $G = \{x_0^1, \dots, x_0^k\}$ and compute the raw reward scores for each generated image: $S_{dp} = \{s_{dp}^0, \dots, s_{dp}^k\}$ for detail preservation and $S_{pf} = \{s_{pf}^0, \dots, s_{pf}^k\}$ for instruction following. We then normalize each dimension

independently:

$$\hat{s}_{dp}^i = \frac{s_{dp}^i - \text{mean}(S_{dp})}{\text{std}(R_{dp})},$$
$$\hat{s}_{pf}^i = \frac{s_{pf}^i - \text{mean}(S_{pf})}{\text{std}(R_{pf})}. \tag{12}$$

This standardization operation maps both reward scores onto a common, dimensionless scale with zero mean and unit variance. Consequently, each normalized score reflects the sample's **relative standing** within the batch rather than its absolute magnitude, enabling fair comparison and balanced optimization across both dimensions.

We then combine the standardized rewards through a weighted sum:

$$s^i = w_{dp} \cdot \hat{s}_{dp}^i + w_{pf} \cdot \hat{s}_{pf}^i, \tag{13}$$

where $w_{dp}$ and $w_{pf}$ are the weights for detail preservation and instruction following, respectively, satisfying $w_{dp} + w_{pf} = 1$. This reward $s_i$ not only measures overall quality but also implicitly enforces balance between image alignment and instruction following.

### 3.3. Negative-aware Score Preference Optimization

As shown in Equation (5), DSPO defines the Preferred Score by decomposing the score function into a pretrained prior and a preference guidance term. The former captures the distribution score over preference data, while the latter is derived from comparisons between annotated preferred and negative preference samples. However, DSPO overlooks the distribution score over negative preference data and requires extensive manual annotation. To address these limitations, we propose NaSPO, which identifies preferred samples through within-group relative advantages, eliminating the need for annotated data. Moreover, we introduce both preferred and negative preference scores to fully leverage information from both preferred and negative preference samples. Please refer to the Appendix A for the algorithm overview of NaSPO.

**Within-group Preference Probability.** As illustrated in Figure 2, for a group of generated images $G = \{x_0^1, \ldots, x_0^k\}$, we compute rewards $R = \{s^0, \ldots, s^k\}$ using our multi-objective reward function. Motivated by DiffusionNFT practices, we transform each raw reward $r^i$ into an optimality probability $Adv_i$:

$$Adv_i = \frac{1}{2} + \frac{1}{2}\text{clip}\left[\frac{s^i - \text{mean}(\{s\}^{1:K})}{\text{std}(\{s\}^{1:K})}, -1, 1\right]. \tag{14}$$

Note that since we use LVLM as the reward model to simulate human judgment, $Adv_i$ can also be interpreted as the probability of human preference. Unlike DSPO's global

preference definition, we determine preference through relative probabilities within each group, which significantly reduces dependency on annotated data.

**Positive Preference Score.** We propose a preferred score that steers the model's optimization toward preferred image distributions. For an image $x_0$ with probability $Adv$ of being preferred, DSPO additionally requires negative images and computes the gradient using the relative margin between a winner and a specific loser ($r(c, x_t^w) - r(c, x_t^l)$), as detailed in Equation (5). In this work, however, inspired by Diffusion-KTO (Li et al., 2024b), we introduce a reference point to shift the focus from absolute utility maximization to relative optimization. Specifically, for a positive image, we aim for the KL divergence between $\pi_\theta(x_t \mid c)$ and $\pi_{\text{old}}(x_t \mid c)$ to be lower than the expected KL divergence over the entire data distribution. Consequently, we obtain $\nabla_{x_t} \log p(y \mid x_t, c) = \nabla_{x_t} \log \sigma(r(c, x_t) - Q_{\text{old}})$, where $Q_{\text{old}} = \beta D_{\text{KL}}[\pi_\theta(x_t \mid c) \| \pi_{\text{old}}(x_t \mid c)]$. Based on this, the Preferred Score is formulated as:

$$v_{t+1}^+ = v_{t+1} + \left(1 - \sigma(r(c, x_t) - Q_{\text{old}})\right)(v_{\theta, t+1} - v_{\text{old}, t+1}). \tag{15}$$

This formulation replaces the dynamic loser term $r(c, x_t^l)$ from DSPO with a reference point $Q_{\text{old}}$, enabling optimization without requiring paired data.

**Negative Preference Score.** While preferred score guides the model toward desirable directions, Negative Preference Score aims to repel the model from undesirable distributions. For image $x_0$ with probability $1 - Adv$ of being undesirable, given a reference point $v_{\text{old}}$, our objective is to move in the opposite direction relative to this reference. We begin by analyzing the data distribution: for an undesirable sample, the goal is not to follow its data-induced velocity $v$, but to diverge from it. We construct a repulsive target via geometric reflection. The vector from the current model prediction to the data velocity is $(v_{t+1} - v_{\text{old}, t+1})$. To repel the model, we invert this direction to $(v_{\text{old}, t+1} - v_{t+1})$ and apply it to the anchor point, yielding:

$$v_{\text{old}, t+1} + (v_{\text{old}, t+1} - v_{t+1}) = 2v_{\text{old}, t+1} - v_{t+1}. \tag{16}$$

Similarly, we aim to steer the velocity field away from the preference direction associated with the negative sample. The gradient direction defined by $x_t$ is $\nabla_{x_t} \log \sigma(Q_{\text{old}} - r(c, x_t))$. Intuitively, this leads to the formulation of the negative preference score:

$$v_{t+1}^- = 2v_{\text{old}} - v_{t+1} - \nabla_{x_t} \log \sigma(Q_{\text{old}} - r(c, x_t)). \tag{17}$$

Upon simplification, we obtain the final formulation for the negative preference score:

$$v_{t+1}^- = v_{t+1} + \left(1 + \sigma(Q_{\text{old}} - r(c, x_t))\right)(v_{\theta, t+1} - v_{\text{old}, t+1}). \tag{18}$$

The detailed derivation is provided in the Appendix C.

**Training Objective.** Similar to DiffusionNFT, we introduce a contradictory loss:

$$\mathcal{L}(\theta) = \mathbb{E}_{c, \pi^{\text{old}}(x_0, c, t)} \Big[ Adv \, \|v_{\theta, t+1} - v_{t+1}^+\|_2^2 \\ + (1 - Adv) \, \|v_{\theta, t+1} - v_{t+1}^-\|_2^2 \Big], \quad (19)$$

where $Adv$ is the preference probability, and $v_{t+1}^+$ and $v_{t+1}^-$ are the positive and negative preference scores, respectively.

Interestingly, we find our NaSPO shares similarities to DiffusionNFT, but effectively substitutes its hyperparameters with our $\sigma(\cdot)$. Please refer to Appendix C for more details.

# 4. Experiment

## 4.1. Experimental Setup

**Dataset and Evaluation Metric.** We conduct experiments on the CHEESE dataset (Sun et al., 2026), a large portrait dataset collected from real-world sources containing approximately 24K authentic portrait images. Images from the same portrait collection are paired and annotated with diverse and complex modification instructions, including model poses and camera parameters. The training set contains approximately 578K triplets, and the test set includes approximately 2K unique identity triplets.

Following CHEESE, we adopt CLIP-I, DINO-I, CLIP-T, Qwen-PF, and Qwen-DP as evaluation metrics, computed using OpenCLIP (ViT-G/14) (Ilharco et al., 2021), DINOv2-Small (Oquab et al., 2023), and Qwen2.5VL-72B (Bai et al., 2025), respectively. Among these, Qwen-PF and Qwen-DP employ LVLM to provide fine-grained assessment of instruction following and detail preservation for generated images, offering more reliable evaluation for PPT tasks.

**Baseline Models.** We compare three categories of learning paradigms for the PPT task: **(1)** Zero-shot (ZS) methods, including IP-Adapter+ (Ye et al., 2023) and EasyRef (Zong et al., 2024); **(2)** SFT methods, including DreamBooth Lora (denotes as DB Lora) (Ruiz et al., 2023) and SCHEESE (Sun et al., 2026); and **(3)** RL methods, including FlowGRPO (Liu et al., 2025a), DiffusionNFT (Zheng et al., 2025), and Uniworld-v2 (Li et al., 2025). For RL methods, we perform post-training on the SFT-trained Kontext model and uniformly employ a multi-objective reward mechanism to adapt them from text-to-image generation to the PPT task.

**Implementation Details.** Our experiments are conducted on FLUX.1-Kontext [Dev] (Batifol et al., 2025), fine-tuned using LoRA ($r = 64, \alpha = 128$) on 8 H800 GPUs. We employ a cold-start phase with a small dataset prior to our post-training stage, with most settings following FlowGRPO (Liu et al., 2025a) and Diffusion-NFT (Zheng et al., 2025). We set the group size $G = 8$ with 48 groups per epoch. We report results obtained after 60 training epochs.

*Table 1.* **Quantitative Comparison Results.** Our method achieves the highest scores on CLIP-T, DP, and PF, demonstrating superior capability in following modification instructions while preserving fine-grained details. The best results are in **boldface**, while the second-best results are underlined.

| Method | Paradigm | CLIP-I | DINO-I | CLIP-T | Qwen-DP | Qwen-PF |
|---|---|---|---|---|---|---|
| IP-Adapter+ | ZS | 0.7940 | 0.6990 | 0.3760 | 0.6590 | 0.5490 |
| EasyRef | ZS | 0.7830 | 0.6870 | 0.3580 | 0.6470 | 0.5450 |
| DB LoRA | SFT | 0.7380 | 0.6770 | 0.3950 | 0.4580 | 0.5790 |
| SCheese | SFT | 0.8390 | 0.7730 | 0.4360 | 0.8550 | 0.7930 |
| Kontext | ZS | 0.8482 | 0.8012 | **0.4417** | 0.7824 | 0.7529 |
| +SFT | SFT | 0.8721 | 0.8163 | 0.4363 | 0.8717 | 0.7928 |
|   +FlowGRPO | RL | 0.8720 | 0.8213 | 0.4364 | 0.8783 | 0.8138 |
|   +DiffusionNFT | RL | **0.8768** | **0.8268** | 0.4400 | 0.8894 | 0.8067 |
|   +Uniworld-V2 | RL | 0.8693 | 0.8174 | 0.4396 | 0.8656 | **0.8271** |
|   **+PortraitRL** | RL | 0.8757 | 0.8239 | **0.4417** | **0.8929** | 0.8260 |

## 4.2. Comparison with Baselines

**Quantitative Results.** Table 1 presents quantitative comparison results between PortraitRL and all baseline methods. Our observations reveal three key findings: **(1)** Compared to zero-shot methods, PortraitRL achieves significant improvements in both PF and DP metrics. **(2)** Compared to SFT methods, PortraitRL demonstrates consistent gains in PF and DP metrics. Notably, when compared to Kontext+SFT (used as our cold-start model), PortraitRL further enhances detail preservation and instruction following capabilities through reinforcement learning. This improvement can be attributed to our multi-objective reward mechanism, which provides fine-grained evaluation of these capabilities, and NaSPO's effective utilization of both positive and negative preference scores. **(3)** Compared to other RL methods, PortraitRL achieves the best performance in terms of the average of DP and PF. Since all four RL methods employ multi-objective rewards, this improvement clearly shows the effectiveness of our NaSPO.

**Qualitative Results.** Figure 3 further presents qualitative comparisons of several RL methods, demonstrating that PortraitRL achieves superior detail preservation and instruction following capabilities. In the first example, PortraitRL successfully preserves the female subject compared to Kontext with SFT, and correctly raises the hands of both people rather than only the male's, better aligning with user preferences. In the seventh example, while DiffusionNFT generates distorted images, PortraitRL produces high-quality images with natural poses. Additional examples further validate our approach: in the fourth example, only our method preserves the wall; in the third and fifth examples, PortraitRL exhibits more natural pose transitions. These results demonstrate that PortraitRL effectively leverages LVLM to simulate user preferences and learns from them through NaSPO. Please refer to Appendix E for more examples.

**User Study.** To evaluate the performance of our method, we conducted a user study with 5 users using a random subset

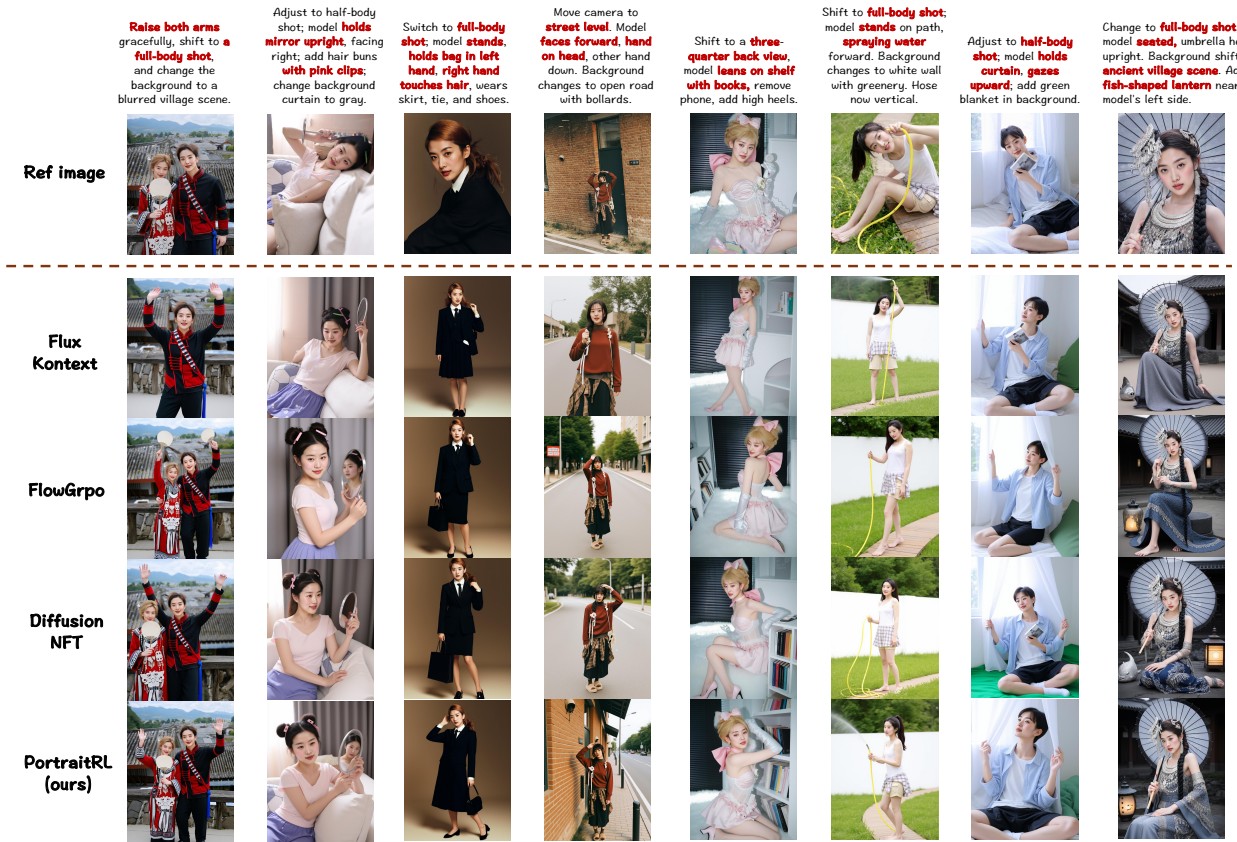

*Figure 3.* **Qualitative Comparisons.** We present a comparison between samples generated by the SFT model (before RL stage) and the outputs from post-training methods, including FlowGRPO, DiffusionNFT, and NaSPO.

*Table 2.* **User Study.** The results mostly align with the LVLM-based scores. **Bold** indicates the best results.

| Method | Human-DP | Human-PF |
|---|---|---|
| +SFT | 0.8767 | 0.7133 |
|   +FlowGRPO | 0.8833 | 0.7883 |
|   +DiffusionNFT | 0.8883 | 0.7817 |
| **+ProtraitRL (Ours)** | **0.9100** | **0.8450** |

*Table 3.* **Ablation Study** on different reward settings. The best results are in **boldface**, while the second-best results are underlined.

| Reward | CLIP-I | DINO-I | CLIP-T | Qwen-DP | Qwen-PF |
|---|---|---|---|---|---|
| - (SFT) | 0.8721 | 0.8163 | 0.4363 | 0.8717 | 0.7928 |
| CLIP-T | **0.8764** | 0.8264 | **0.4423** | 0.8813 | 0.7953 |
| CLIP-I | 0.8735 | 0.8248 | 0.4362 | 0.8746 | 0.8108 |
| DINO-I | 0.8736 | **0.8272** | 0.4361 | 0.8710 | 0.8060 |
| Dual-PD | 0.8684 | 0.8161 | 0.4417 | 0.8697 | **0.8357** |
| **Ours** | 0.8757 | 0.8239 | 0.4417 | **0.8929** | 0.8260 |

of 50 samples from the CHEESE test set. Users independently scored each image on a scale of 0 to 4 based on these two criteria, i.e., DP and PF, respectively. The results are presented in Table 2. As shown in the table, images generated by our method achieved the highest scores in both DP and PF, demonstrating its capability to simultaneously enhance performance in both Prompt Following and Detail Preservation. Note that the user study results align with the LVLM-based scores in Table 1, further validating the effectiveness of using LVLM for evaluation.

### 4.3. Ablation Study

**Ablation on Different Rewards Setting.** To validate the effectiveness of our proposed reward mechanism, we conduct a comprehensive ablation study comparing various reward

formulations listed in Table 3. Firstly, utilizing a standard encoder-based reward to combine image and text similarity ($R = S_{\text{CLIP}}(I_{\text{gen}}, I_{\text{tgt}}) + S_{\text{CLIP}}(I_{\text{gen}}, T)$) increased CLIP scores but underperformed on LVLM benchmarks, revealing the limitations of CLIP-T in ensuring effective prompt adherence. Secondly, to address this, we substituted CLIP-T with a directional metric $[S_{\text{CLIP}}(I_{\text{gen}}, I_{\text{tgt}}) - S_{\text{CLIP}}(I_{\text{gen}}, I_{\text{ref}})]$ and extended this approach to DINO to better capture local features. However, the continued reliance on target images restricted the method's generalization capabilities. Thirdly, we tasked an LVLM with a unified instruction to evaluate both aspects simultaneously. While prompt following improved, detail preservation suffered, indicating that the

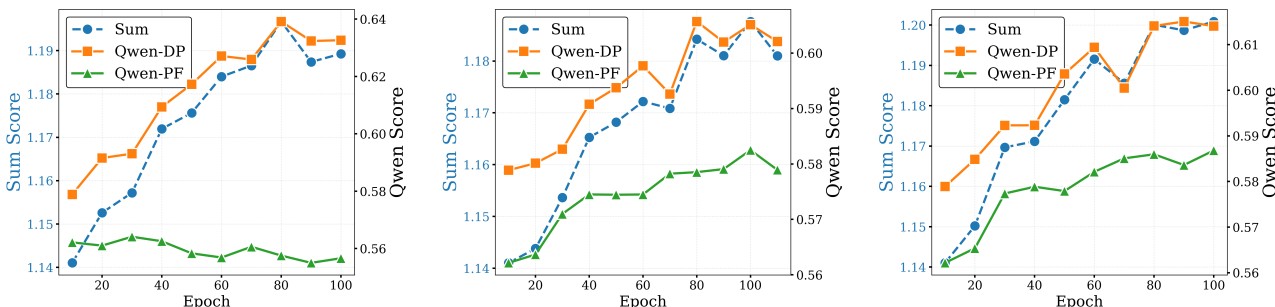

*Figure 4.* **Reward Hacking Phenomenon**. Orange and green curves correspond to Qwen-DP and Qwen-PF scores, with the blue curve indicating their aggregate sum. The figures compare three aggregation strategies: without normalization (left), normalization across the epoch dimension (middle), and normalization across the group dimension (right).

*Table 4.* **Ablation Study** on different standardization schemes. The best results are in **boldface**.

| Method | Qwen-DP | Qwen-PF |
|---|---|---|
| w/o Standardization | 0.8854 | 0.7899 |
| Within-epoch Std. | 0.8917 | 0.8125 |
| **Within-group Std. (Ours)** | **0.8926** | **0.8260** |

LVLM struggles to balance the trade-off between these conflicting objectives under a single directive. Consequently, our final approach decouples these metrics using a two-stage normalization strategy, successfully enhancing performance in both dimensions.

**Impact of Reward Normalization Strategies.** We investigate the efficacy of normalization across epoch and group dimensions compared to a baseline without normalization. As shown in Table 4, we observe that the absence of normalization leads to poor performance in prompt following evaluation. Conversely, both normalization strategies successfully suppress reward hacking, with group-wise normalization proving to be more training-efficient and yielding superior overall performance.

### 4.4. Further Analysis

**Reward Hacking.** We investigate how our Multi-Objective Reward Mechanism works in our work. As visualized in Figure 4, employing a naive additive strategy, which means directly summing the scores results in a rapid initial increase in total reward. However, a closer inspection reveals that this growth is disproportionately driven by the Detail Preservation score, while the model's adherence to textual instructions degrades. For PPT, model may by copying the reference image to gain a higher DP score. This suggests the model exploits the reward landscape by prioritizing the easier objective at the cost of the harder one. Quantitatively, as shown in Table 3, this imbalance leads to a performance regression in the Qwen-PF metric compared to the baseline. In contrast, applying normalization along either the epoch or group dimension prior to summation effectively mitigates this issue, resulting in more stable reward growth during

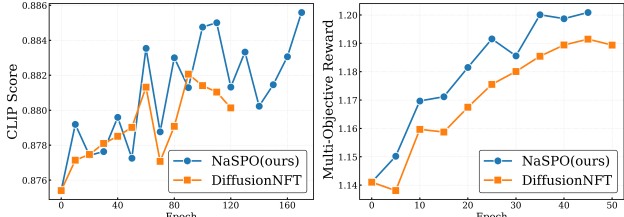

*Figure 5.* **Comparison of Training Reward Trajectories between NaSPO and DiffusionNFT.** The sub-figures display the curves for CLIP Reward (left), and Multi-Objective Reward (right).

training. By functioning as a pseudo-dynamic weighting mechanism, this strategy ensures simultaneous improvements in both objectives. Consequently, within-group standardization achieves superior convergence and better alignment with the dual goals of preservation of image details and precise instruction following.

**Comparison of Training Dynamics.** To further compare the performance of NaSPO and DiffusionNFT, we tracked the training reward trajectories across different metrics. As show in Figure 5, NaSPO consistently outperforms the DiffusionNFT under identical training settings. This superiority demonstrates that our optimization strategy is both robust and effective in maximizing the target objectives.

## 5. Conclusion

In this paper, we propose PortraitRL, a novel RL framework tailored for the inherent challenges in portrait pose transfer. Our approach consists of two key components: **(1)** a Multi-Objective Reward Mechanism with two LVLM logit-based reward functions that simulate human judgment for fine-grained evaluation of prompt following and detail preservation; **(2)** Negative-aware Score Preference Optimization (NaSPO), a novel RL algorithm that models human preferences through within-group relative advantages and defines both positive and negative preference scores to fully leverage information from all generated samples. Extensive experiments demonstrate that our method achieves state-of-the-art performance, yielding significant improvements in both identity preservation and pose accuracy.

## Impact Statement

This paper aims to advance the field of Portrait Pose Transfer. Similar to other image editing tasks, our work has potential applications across various domains. While we acknowledge the general societal implications of generative models, none which we feel must be specifically highlighted here.

## Acknowledgements

This work is partially supported by National Natural Science Foundation of China (62376274, 62437002). Zhiwu Lu is the corresponding author.

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

# A. Algorithm for NaSPO

---

**Algorithm 1** Negative-aware Score Preference Optimization (NaSPO)

---

**Require:** Pretrained diffusion policy $v^{\text{ref}}$, raw reward function $r^{\text{raw}}(\cdot) \in \mathbb{R}$, prompt dataset $\{c\}$.
  **Initialize:** Data collection policy $v^{\text{old}} \leftarrow v^{\text{ref}}$, training policy $v_\theta \leftarrow v^{\text{ref}}$, data buffer $\mathcal{D} \leftarrow \emptyset$.
  **for** each iteration $i$ **do**
    **for** each sampled prompt $c$ **do**
      Sample trajectories $\{x_0^{(k)}\}_{k=1}^K \sim v_{\text{old}}(\cdot|c)$.
      Evaluate metrics: $s_{\text{dp}}, s_{\text{pf}} \in \mathbb{R}^K$.
      **Two-stage Reward Normalization:**
        $\hat{s}^i \leftarrow \text{Norm}(\{s_{\text{dp}}\}^{1:K}) + \text{Norm}(\{s_{\text{pf}}\}^{1:K})$
        $s^i \leftarrow \text{Norm}(\{\hat{s}\}^{1:K})$
      Calculate advantage: $Adv^{(i)} \leftarrow s^i + 0.5$ and update $\mathcal{D}$.
    **end for**
    **for** each mini batch $c, x_0, r \in D$ do **do**
      Forward diffusion process: $x_t = \alpha_t x_0 + \sigma_t \epsilon$; $v = \dot{\alpha}_t x_0 + \beta_t \epsilon$.
      Compute Positive Preference Score.
      Compute Negative Preference Score.
      $\theta \leftarrow \theta - \lambda \nabla_\theta$
    **end for**
    Update data collection policy $\theta^{\text{old}} \leftarrow \eta_i \theta^{\text{old}} + (1 - \eta_i)\theta$, and clear buffer $\mathcal{D} \leftarrow \emptyset$.
  **end for**
  **Output:** $v_\theta$

---

# B. Theoretical Discussions

## B.1. Intution behind the normalization mechanism

Critically, our projection operates on **Z-score normalized** latent vectors rather than raw scores. This step introduces two vital statistical properties that enforce balance and fairness:

Normalization serves as an adaptive weighting mechanism that calibrates the contribution of each metric inversely to its volatility:

$$\tilde{r}_i = \frac{r_i - \mu_i}{\sigma_i} \implies \frac{\partial \tilde{r}_i}{\partial r_i} \propto \frac{1}{\sigma_i}. \tag{20}$$

We posit that dimensions exhibiting high variance ($\sigma_{\text{high}}$) often imply a lower "cost" of achieving high magnitude scores. In *portrait pose transfer*, the image-alignment metric ($r_x$) is susceptible to trivial shortcuts (e.g., copying), leading to high volatility. By scaling by $1/\sigma_x$, we dampen rewards for such "cheap" gains while amplifying signals from the harder, more stable instruction-following metric.

## B.2. Discussion on the Summation of Normalized Rewards

It is well established that the weighted summation of rewards is fundamentally equivalent to projecting the reward vector $v = (r_x, r_y)$ onto a target direction vector $d = (1, 1)$. This operation acts as a computationally efficient proxy for minimizing the distance to an idealized state. However, this approach may harbor certain limitations. For instance, in a scenario with weights $(1, 1)$, the normalized vectors $(2, 2)$ and $(1, 3)$ yield an identical scalar reward, yet a balanced solution such as $(2, 2)$ is typically preferred over an imbalanced one like $(1, 3)$.

Prior literature has hypothesized the existence of a model-dependent theoretical maximum, or Utopia point $z = (\lambda, \lambda)$, and utilized the Euclidean distance between samples and this point as a ranking metric. We investigated this approach empirically. Specifically, we defined the Utopia point dynamically using the maximum values observed within an epoch or a group. Through our experiments, we find that while this method yields performance improvements under within-epoch standardization, it leads to performance degradation under within-group standardization. We hypothesize that the Utopia point functions as an adaptive hyperparameter that is highly sensitive to batch size. In within-group standardization,

normalization is performed over the entire epoch, providing a larger sample size that stabilizes this adaptive hyperparameter. Conversely, the smaller sample size in the group setting introduces instability. We suggest that appropriately increasing the number of groups could potentially mitigate this issue, which remains a subject for our future investigation.

$$\mathcal{L}(v) = \|z^* - v\|_2^2$$
$$= 2\lambda^2 - 2\lambda(r_x + r_y) + (r_x^2 + r_y^2). \tag{21}$$

As the standard for perfection becomes arbitrarily high ($\lambda \to \infty$), the linear interaction term $2\lambda(r_x + r_y)$ dominates the quadratic penalty term $(r_x^2 + r_y^2)$. Consequently, minimizing the distance to the Utopia point becomes mathematically equivalent to maximizing the projection sum:

$$\arg\min_v \|z^* - v\|_2^2 \approx \arg\max_v (r_x + r_y) \quad \text{as } \lambda \to \infty. \tag{22}$$

This derivation justifies the use of linear projection. Consequently, in this work, we adopt a straightforward weighted summation approach.

## C. Derivation of Section 3.3

### C.1. Derivations for Equation (6)

Following the formulation proposed in (Liu et al., 2025a), the score function for the flow matching process is expressed as:

$$\nabla \log p_t(x) = -\frac{x}{t} - \frac{1-t}{t} v_t(x), \tag{23}$$

where $x$ and $v_t$ denote the state and the velocity vector, respectively. Consequently, the score difference between the model distribution $p_\theta$ and the data distribution $p_{\text{data}}$ can be derived as follows:

$$\nabla_{x_t} \log \frac{p_\theta}{p_{\text{data}}} = \left(-\frac{x_t}{t} - \frac{1-t}{t} v_\theta(x_t, c)\right) - \left(-\frac{x_t}{t} - \frac{1-t}{t} v_{\text{data}}(x_t, c)\right)$$
$$= -\frac{1-t}{t} v_\theta(x_t, c) + \frac{1-t}{t} v_{\text{data}}(x_t, c) \tag{24}$$
$$= -\frac{1-t}{t} \left(v_\theta(x_t, c) - v_{\text{data}}(x_t, c)\right).$$

Following (Liu et al., 2025b), we obtain the definition of the reward gradient, which is given by:

$$\nabla_{x_t} r(c, x_t) = \lambda \log \frac{p_\theta(x_t \mid x_{t+1}, c)}{p_{\text{ref}}(x_t \mid x_{t+1}, c)} \tag{25}$$
$$= -\beta_t \left(v_\theta(x_{t+1}, c, t+1) - v_{\text{ref}}(x_{t+1}, c, t+1)\right),$$

where $\beta_t = \beta(1-t)^2$. By substituting these results into Equation (3), we derive the DSPO score specifically adapted for the flow matching framework. This formulation connects the velocity difference directly to the preference signal:

$$\frac{1}{\gamma} \nabla_{x_t} \log \frac{p_\theta}{p_{\text{data}}} = \nabla_{x_t} \log p(y \mid x_t, c)$$
$$A(t)\left(v_\theta(x_t, c) - v_{\text{data}}(x_t, c)\right) = \left(1 - \sigma(r(c, x_t) - r(c, x_t^l))\right) \nabla_{x_t} r(c, x_t) \tag{26}$$
$$A(t)\left(v_\theta(x_t, c) - v_{\text{data}}(x_t, c)\right) = \left(1 - \sigma(r(c, x_t) - r(c, x_t^l))\right)$$
$$\cdot \left(v_\theta(x_{t+1}, c, t+1) - v_{\text{ref}}(x_{t+1}, c, t+1)\right).$$

where the time-dependent coefficient is defined as $A(t) = \frac{1-t}{t\gamma\beta_t}$. In practice, based on empirical findings and to further simplify the optimization objective, we omit the weighting term $A(t)$ in our experiments.

### C.2. Derivations of Equation (15)

For positive samples, the positive preference score is derived from:

$$\nabla_{x_t} \log p_\theta(x_t \mid c, y_{\text{pref}}) = \nabla_{x_t} \log p(x_t \mid c) + \gamma \nabla_{x_t} \log p(y_{\text{pref}} \mid x_t, c). \tag{27}$$

This leads to the following formulation:

$$v_{\theta,t+1} = v_{t+1} + \nabla_{x_t} \log \sigma(r(c, x_t)) \tag{28}$$

$$= v_{t+1} + (1 - \sigma(r(c, x_t)))\nabla_{x_t} r(c, x_t) \tag{29}$$

$$= v_{t+1} + (1 - \sigma(r(c, x_t)))(v_{\theta,t+1} - v_{\text{old},t+1}). \tag{30}$$

## C.3. Derivations of Equation (18)

For the negative preference score, we aim to express the same optimization objective using the negative sample probability $p(y = \text{negative} \mid x_t, c)$. Starting from the identity:

$$\nabla_{x_t} \log p(y_{\text{pref}} \mid x_t, c) = \nabla_{x_t} \log(1 - p(y_{\text{non-pref}} \mid x_t, c))$$

$$= -\frac{\nabla_{x_t} p(y_{\text{non-pref}} \mid x_t, c)}{1 - p(y_{\text{non-pref}} \mid x_t, c)}. \tag{31}$$

Since $\nabla_{x_t} \log p(y_{\text{non-pref}} \mid x_t, c) = \frac{\nabla_{x_t} p(y_{\text{non-pref}} \mid x_t, c)}{p(y_{\text{non-pref}} \mid x_t, c)}$, we can rewrite the gradient as:

$$\nabla_{x_t} \log p(y_{\text{pref}} \mid x_t, c) = -w \nabla_{x_t} \log p(y_{\text{non-pref}} \mid x_t, c). \tag{32}$$

while the weighting coefficient $w = \frac{p(y_{\text{non-pref}} \mid x_t, c)}{1 - p(y_{\text{non-pref}} \mid x_t, c)}$. This formulation naturally leads to the negative preference score:

$$\nabla_{x_t} \log p_\theta(x_t \mid c, y_{\text{pref}}) = \nabla_{x_t} \log p(x_t \mid c) - \gamma w \nabla_{x_t} \log p(y_{\text{non-pref}} \mid x_t, c). \tag{33}$$

From the above derivation, we can see that the positive and negative preference scores form a contrastive pair of optimization directions, which improves the stability of our online reinforcement learning process. Expanding the formulation in Equation (33):

$$v_{\theta,t+1} = (2v_{\text{old},t+1} - v_{t+1}) - \nabla_{x_t} \log \sigma(-r(c, x_t)) \tag{34}$$

$$= 2v_{\text{old},t+1} - v_{t+1} + (1 - \sigma(-r(c, x_t)))(\nabla_{x_t} r(c, x_t)) \tag{35}$$

$$= 2v_{\text{old},t+1} - v_{t+1} + (1 - \sigma(-r(c, x_t)))(v_{\theta,t+1} - v_{\text{old},t+1}). \tag{36}$$

Subtracting $2v_\theta$ from both sides to regroup the terms:

$$-v_{\theta,t+1} = (2v_{\text{old},t+1} - 2v_{\theta,t+1}) - v_{t+1} + (1 - \sigma(-r(c, x_t)))(v_{\theta,t+1} - v_{\text{old},t+1}) \tag{37}$$

$$= -2(v_{\theta,t+1} - v_{\text{old},t+1}) - v_{t+1} + (1 - \sigma(-r(c, x_t)))(v_{\theta,t+1} - v_{\text{old},t+1}) \tag{38}$$

$$= -v_{t+1} + [1 - \sigma(-r(c, x_t)) - 2](v_{\theta,t+1} - v_{\text{old},t+1}). \tag{39}$$

Since $1 - \sigma - 2 = -(1 + \sigma)$, we simplify the expression to:

$$-v_{\theta,t+1} = -v_{t+1} - (1 + \sigma(-r(c, x_t)))(v_{\theta,t+1} - v_{\text{old},t+1}). \tag{40}$$

Multiplying by $-1$, we obtain the final update rule:

$$v_{\theta,t+1} = v_{t+1} + (1 + \sigma(-r(c, x_t)))(v_{\theta,t+1} - v_{\text{old},t+1}). \tag{41}$$

## C.4. Derivation and Analysis of the Closed-Form Solution

For positive samples, deriving from Eq. (28), the solution is obtained as follows:

$$v_{\theta,t+1} = v_{t+1} + (1 - \sigma(r(c, x_t)))(v_{\theta,t+1} - v_{\text{old},t+1})$$

$$\sigma(r(c, x_t))v_{\theta,t+1} = v_{t+1} - (1 - \sigma(r(c, x_t)))v_{\text{old},t+1}$$

$$\sigma(r(c, x_t))v_{\theta,t+1} = v_{t+1} - v_{\text{old},t+1} + \sigma(r(c, x_t))v_{\text{old},t+1}$$

$$v_{\theta,t+1} = v_{\text{old},t+1} + \frac{1}{\sigma(r(c, x_t))}(v_{t+1} - v_{\text{old},t+1}). \tag{42}$$

*Table 5.* Training settings.

| HYPERPARAMETERS | VALUE | HYPERPARAMETERS | VALUE |
|---|---|---|---|
| **LoRA:** | | **Reward:** | |
| $r$ | 64 | $w_{dp}$ | 1.0 |
| $\alpha$ | 128 | $w_{pf}$ | 1.0 |
| **Basic:** | | **Sampling:** | |
| LEARINING RATE | 2E-4 | SAMPLING INFERENCE STEPS | 25 |
| $\beta_1$ | 0.9 | RESOLUTION | $512 \times 512$ |
| $\beta_2$ | 0.999 | THE NUMBER OF IMAGES PER PROMPT | 8 |
| BATCH SIZE | 1 | THE NUMBER OF GROUPS | 48 |
| EMA DECAY | 0.9 | | |

*Table 6.* Results trained with different hyperparameters $w_{dp}$ and $w_{pf}$.

| PROPORTION OF $w_{dp} : w_{pf}$ | CLIP-I | DINO-I | CLIP-T | QWEN-DP | QWEN-PF | QWEN-AVG |
|---|---|---|---|---|---|---|
| 1.0:1.5 | 0.8747 | 0.8251 | 0.4420 | 0.8897 | 0.8269 | 0.8583 |
| 1.5:1.0 | 0.8807 | 0.8300 | 0.4389 | 0.8958 | 0.7983 | 0.8471 |
| 1.0:1.0 | 0.8760 | 0.8250 | 0.4417 | 0.8953 | 0.8280 | **0.8617** |

For better representation, this can be written as:

$$v_{t+1}^+ = v_{\text{old},t+1} + \frac{1}{\sigma(r(c, x_t))}(v_{t+1} - v_{\text{old},t+1}). \tag{43}$$

Similarly, for negative samples, the solution is given by:

$$v_{t+1}^- = v_{\text{old},t+1} - \frac{1}{\sigma(-r(c, x_t))}(v_{t+1} - v_{\text{old},t+1}). \tag{44}$$

In the specific case of DiffusionNFT, the optimal solution satisfies the following conditions:

$$v_{t+1}^+ = v_{\text{old},t+1} + \frac{1}{\beta}(v_{t+1} - v_{\text{old},t+1}), \tag{45}$$

$$v_{t+1}^- = v_{\text{old},t+1} - \frac{1}{\beta}(v_{t+1} - v_{\text{old},t+1}). \tag{46}$$

It is evident that this formulation is formally equivalent to the general solution derived in Eq. (42) and Eq. (44). The primary distinction lies in the substitution of the adaptive term $\sigma(r(c, x_t))$ with the constant hyperparameter $\beta$. Thus, in its final mathematical form, our method essentially represents an incremental advancement upon Dif fusionNFT.

## D. Experimental Setup

### D.1. Training Configuration

Our training configurations largely adhere to those used in FlowGRPO and DiffusionNFT. Due to GPU memory constraints, we set the batch size to 1, resulting in a group size of 8. We utilize a LoRA configuration with $\alpha = 128$ and $r = 64$, and a learning rate of $2 \times 10^{-4}$. To ensure consistency, the sampling inference steps are set to 25, at a resolution of $512 \times 512$. As shown in Table 5, we provide all the hyperparameters we used in our paper.

### D.2. Reward Design and Hyperparameter Study

As shown in Table 6, we present results trained with different hyperparameters $w_{dp}$ and $w_{pf}$. We can see that increasing the weight of either $w_{dp}$ or $w_{pf}$ leads to a moderate decrease in the score of the other metric.

### D.3. Computational Complexity

Additionally, we provide a detailed comparison of computational complexity in Table 7. As shown, all three methods share the same trainable parameter ratio (1.4414%), indicating that PortraitRL introduces no additional trainable parameters.

*Table 7.* Computational complexity.

| METHOD | TRAINABLE PARAM% | PEAK MEM | SAMPLE TIMES | OPTIM TIMES | THROUGHPUT |
|---|---|---|---|---|---|
| FLOWGRPO | 1.4414% | 47.61 GB | 168.85s | 379.23s | 0.70 |
| DIFFUSIONNFT | 1.4414% | 47.95 GB | 190.30s | 665.46s | 0.45 |
| PORTRAITRL | 1.4414% | 47.95 GB | 190.82s | 662.98s | 0.45 |

*Table 8.* Zero-shot Results on ImgEdit.

| TASK | ACTION | ADJUST | STYLE | BACKGROUND | EXTRACT | REMOVE | REPLACE | ADD | COMPOSE | AVG |
|---|---|---|---|---|---|---|---|---|---|---|
| FLOWGRPO | 0.7833 | 0.8894 | 0.7740 | 0.9383 | **0.5265** | **0.6930** | 0.9500 | 0.9537 | **0.7043** | 0.7980 |
| DIFFUSIONNFT | 0.8611 | 0.8723 | **0.8040** | 0.9426 | 0.4838 | 0.5744 | 0.9478 | 0.9411 | 0.6522 | 0.7772 |
| PORTRAITRL | **0.9222** | **0.9000** | **0.8040** | **0.9766** | 0.5179 | 0.6116 | **0.9587** | **0.9663** | 0.6609 | **0.8074** |

*Table 9.* GPT-4o-based evaluation Results.

| METHOD | QWEN-DP | QWEN-PF | GPT-DP | GPT-PF |
|---|---|---|---|---|
| KONTEXT+SFT | 0.8717 | 0.7928 | 0.8915 | 0.8515 |
| +FLOWGRPO | 0.8783 | 0.8138 | 0.8955 | 0.8625 |
| +DIFFUSIONNFT | 0.8894 | 0.8067 | 0.9040 | 0.8590 |
| +PORTRAITRL | **0.8929** | **0.8260** | **0.9210** | **0.8800** |

Moreover, PortraitRL exhibits nearly identical peak memory, sampling time, optimization time, and throughput compared to DiffusionNFT, demonstrating that our proposed NaSPO method introduce negligible computational overhead.

### D.4. Prompt Template

In Figure 6, we present the prompt templates used for evaluating Detail Preservation (DP) and Prompt Following (PF) during the reward function and evaluation phases, as well as the supplementary Dual Prompt & Detail Evaluation (Dual-PD) templates utilized in the ablation study.

## E. Additional Results

### E.1. Comparison on ImgEdit

Beyond portrait photo transformation, PortraitRL also demonstrates strong performance on general editing tasks. As shown in Table 8, we report zero-shot comparisons between PortraitRL and other baselines on the ImgEdit dataset. The results showed that PortraitsRL still achieved the highest average score.

### E.2. GPT-4o-Based Evaluation

When GPT-4o is used instead of Qwen-VL for evaluation, the obtained new results (in terms of GPT-DP and GPT-PF) are presented in Table 9. It shows consistent conclusions, no matter which LVLM is used for evaluation.

### E.3. Qualitative Results

Additional qualitative results are provided in Figure 7. The visual examples in these supplementary comparisons corroborate our earlier findings from Figure 3, consistently showing that our method achieves superior detail preservation and more accurate prompt following compared to the baselines.

## Template for Detail Preservation Evaluation

**System Prompt:** I understand this task involves evaluating the detail consistency between a reference photo and a generated photo to determine if they could be part of the same portrait collection.
The evaluation focuses on four main aspects: model details, outfit details, photography style, and technical quality.
The evaluation should result in a specific score ranging from 0 (completely inconsistent) to 4 (nearly identical).
*** I will strongly penalize any instances where the generated photo is directly copied from the reference photo by assigning a score of 0.
The evaluation will follow these steps:
1. Model Details: Assess if the generated image maintains consistency in facial features (shape of features, skin texture), makeup (lipstick, eye makeup, etc.), and hairstyle/color with the reference image.
2. Outfit Details: Examine the completeness of clothing and accessories, including garment style, fabric texture, draping effects, and accessory details to ensure they are identical to the reference image.
3. Photography Style: Compare lighting effects, color grading, background style between the two photos to ensure they align with a cohesive portrait collection.
4. Technical Quality: Evaluate the generated image's sharpness, exposure accuracy, color reproduction, and check for any distortions or unnatural proportions.
After analyzing these aspects, I will assign a score based on the overall performance of the generated image in relation to the reference image, assigning a score of 0 in any case where direct copying is evident without added any new changes. The score will reflect how similar the generated image is to the reference, strictly adhering to the evaluation criteria provided, and strongly penalizing direct copying by assigning a 0 score.
My output format should be Score:[0-4], and I don't need to write out the specific analysis process.

**User Prompt:** ### Task Definition
You will be provided with an image generated based on a reference photo. As an experienced evaluator, your task is to assess the detail consistency between the generated photo and the reference photo to determine if they could believably be part of the same photo shoot.
Please penalize any instances where the generated photo is directly copied from the reference photo by assigning a score of 0.
### Evaluation Criteria
The evaluation will be based on four key aspects:
1. Model Details:-Facial feature consistency (shape and proportion of features) -Skin texture and tone -Makeup details (lipstick color, eye makeup style, etc.) -Hair style and color -Distinguishing marks (moles, freckles, etc.)
2. Outfit Details:-Identical clothing items -Fabric texture and patterns -Clothing drape and wrinkles -Accessory details (jewelry, bags, etc.) -Color accuracy
3. Photography Style:-Lighting setup and effects -Color grading treatment -Background style/setting -Depth of field/focus effects
4. Technical Quality:-Image sharpness and clarity -Exposure accuracy -Color reproduction -No obvious artifacts or distortions -Natural body proportions
### Scoring Range
Please provide an integer score from 0-4 based on the overall performance across these features:
-Very Poor (0): Completely inconsistent, cannot be part of the same portrait collection, or it directly copy from the reference image
-Poor (1): Significant differences, difficult to use in the same portrait collection. Also, use this score if the image shows copyied certain components without added new elements or understanding
-Fair (2): Basic similarity but with notable inconsistencies
-Good (3): Highly similar with only minor differences
-Excellent (4): Nearly perfect, can perfectly fit in the same portrait collection
### Input Format
You will receive two images each time, the first being the reference photo and the second being the generated photo. Please carefully examine the details in each photo.
### Output Format:
Directly output the score number (0-4) only.

## Template for Prompt Following Evaluation

**System Prompt:** I understand this task involves evaluating whether a generated image accurately implements the modifications specified in the instruction text when compared to the reference image. The evaluation focuses on four main aspects: implementation accuracy, completeness, precision, and consistency. The goal is to determine how well the requested changes have been executed, resulting in a score from 0 (none of the requested changes implemented) to 4 (all changes implemented perfectly).
To evaluate the modification implementation, I will:
1. Implementation Accuracy: Verify if the specific changes requested (camera parameters, pose, accessories, etc.) have been correctly executed.
2. Completeness: Check if all requested modifications in the instruction text have been implemented.
3. Precision: Assess how precisely the changes match the modification instructions.
4. Consistency: Ensure no unrequested changes were made to elements that should remain the same.
After analyzing these aspects, I will assign a score based solely on how well the requested modifications were implemented, regardless of image quality unless it prevents verification of the changes.

**User Prompt:** ### Task Definition
You will be provided with a reference image, modification instructions, and a generated image. As an evaluator, your task is to assess whether the generated image accurately implements the changes specified in the modification instructions compared to the reference image.
### Evaluation Criteria
The evaluation focuses on how well the requested modifications have been implemented:
1.Implementation Accuracy:-Camera parameter changes (distance, angle) correctly applied -Model pose modifications accurately executed -Accessory/prop changes properly implemented -Other specific modifications correctly executed
2.Completeness: -All requested modifications are present -No missing changes from the instruction text -Full implementation of each modification request -All aspects of complex changes addressed
3.Precision: -Modifications match the exact specifications -Changes implemented to the degree specified -Accurate interpretation of modification instructions -Proper execution of detailed requirements
4.Consistency: -No unrequested changes to other elements -Maintenance of unmodified aspects -No unexpected alterations
### Scoring Range
Please provide an integer score from 0-4 based on modification implementation:
-Very Poor (0): None of the requested changes implemented correctly
-Poor (1): Some changes attempted but mostly incorrect or incomplete
-Fair (2): Major changes implemented but with significant deviations
-Good (3): Most changes implemented correctly with minor deviations
-Excellent (4): All requested changes implemented perfectly
### Input Format
You will receive: A reference image, Modification instructions and A generated image
Please carefully compare the changes between the reference and generated images against the modification instructions.
### Output Format:
Directly output the score number (0-4) only.

## Template for Dual Prompt & Detail Evaluation

**System Prompt:** I understand this task involves evaluating a generated image against a reference image based on modification instructions. The evaluation assesses the balance between **executing requested changes** and **preserving original quality/details**.
The goal is to assign a score (0-4) based on two main pillars:
1. Modification Execution: Accuracy, completeness, and precision of specific requests (pose, camera, props).
2. Consistency & Quality: Preservation of the model's identity, outfit details, photography style, AND overall technical integrity (sharpness, anatomy).
I will prioritize images that accurately implement changes while maintaining the reference's distinct features and high visual quality without introducing artifacts.

**User Prompt:** ### Task Definition
Compare the generated image with the reference image and instructions. Evaluate if the requested changes were implemented accurately while maintaining the reference's identity, style, and technical quality.
### Evaluation Criteria
1. Modification Adherence (Active Changes):- **Accuracy:** Camera/pose/prop changes match instructions exactly. - **Completeness:** All requested modifications are present; none ignored.
2. Consistency & Quality Preservation (Passive Retention):- **Identity & Outfit:** Retains facial features, skin texture, makeup, hair, fabric details, and unrequested accessories. - **Style & Atmosphere:** Maintains original lighting, color grading, and background setting. - **Technical Integrity:** High sharpness/exposure; natural anatomy (no distortions); no artifacts.
### Scoring Range
Output an integer (0-4) based on the success of changes AND quality of preservation:
- **0 (Very Poor):** Changes not implemented, or image is broken/low quality.
- **1 (Poor):** Changes attempted but incorrect; OR severe loss of identity/realism.
- **2 (Fair):** Major changes implemented but with noticeable loss of details, style, or quality.
- **3 (Good):** Changes correct with only minor deviations in consistency or technical quality.
- **4 (Excellent):** Perfect implementation of changes with flawless preservation of identity, details, and technical quality.
### Input Format
Reference image, Modification instructions, Generated image.
### Output Format
Directly output the score number (0-4) only.

*Figure 6.* **Prompt templates** used for Detail Preservation, Prompt Following, and Dual Prompt & Detail Evaluation.

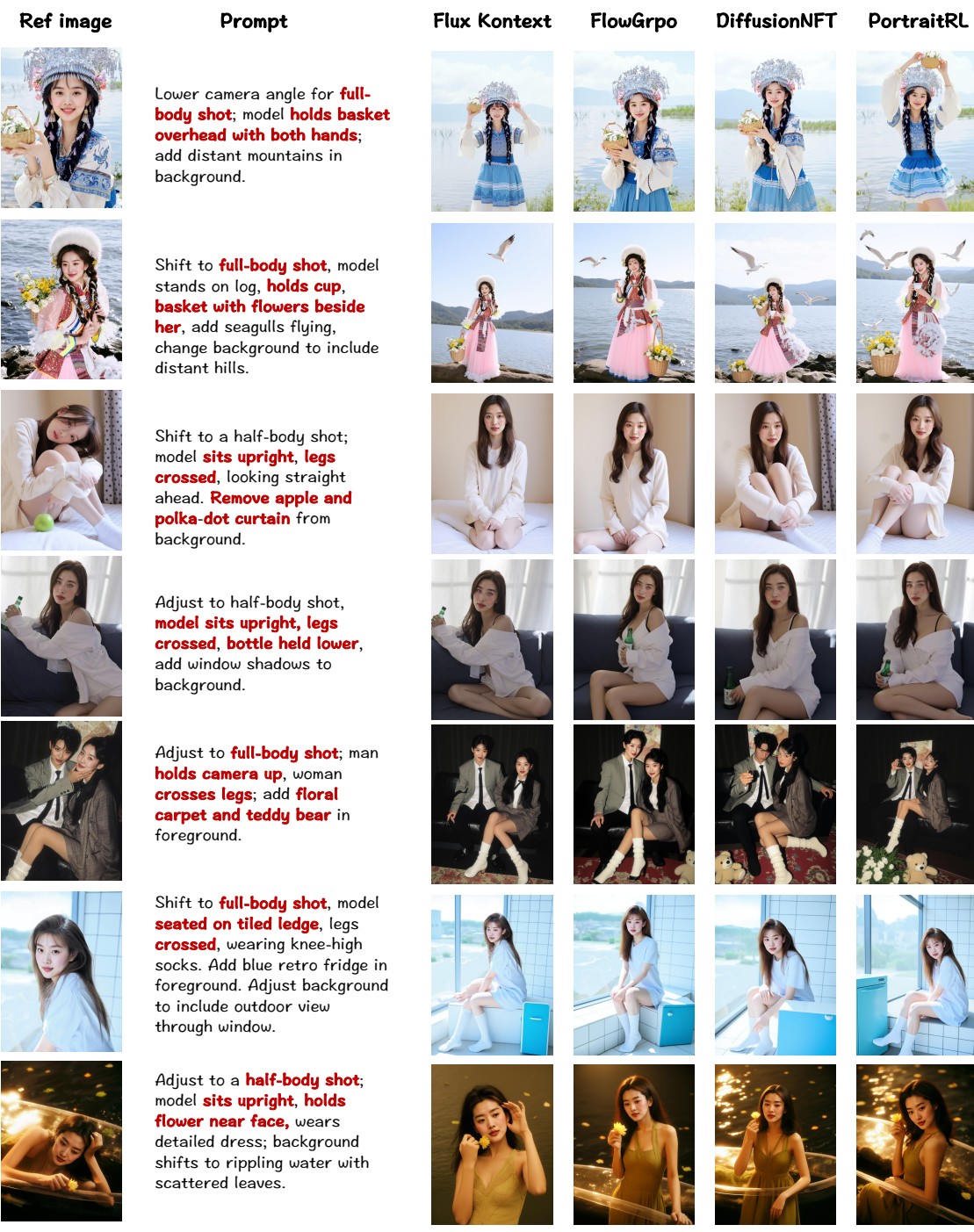

*Figure 7.* **Additional Qualitative Results.** We present more comparisons between samples generated by the SFT model (before RL stage) and the outputs from post-training methods, including FlowGRPO, DiffusionNFT, and NaSPO.

