# OpenReview forum: "PortraitRL: Reinforcement Learning for Personalized Portrait Pose Transfer with Multi-Objective Reward Modeling"
_ICML.cc/2026/Conference — ICML 2026 regular_

### Official Review · Reviewer_mvmP · 2026-03-12

**Soundness:** 3
**Presentation:** 4
**Significance:** 4
**Originality:** 4
**Overall Recommendation:** 5
**Confidence:** 3

**Summary:**

The paper proposes PortraitRL, a post-training framework for PPT which utilizes LVLM-based reward functions to help ensure it follows the prompt and preserves detail. It also proposes NaSPO, a GRPO-based RL algorithm which innovates on DSPO and defines a negative preference score in addition to the positive preference score; with NaSPO, needs for annotation can be greatly reduced. The paper tests the efficacy of the algorithm on the CHEESE dataset and obtains favorable results with respect to identity preservation and pose accuracy.

**Compliance With Llm Reviewing Policy:**

Affirmed.

**Final Justification:**

Overall, I began with a relatively positive impression of the paper. I don't believe I am sufficiently familiar with existing literature to comment at length on the originality of the work. I can, however, state the the paper is clear, addresses a significant problem, and while there are no formal proofs, the claims are decently justified through experiments. I also appreciated that the rebuttal gave additional justification for the methods used. My opinions have not changed greatly over the rebuttal process, and I maintain my initial score.

**Key Questions For Authors:**

1. Is there a reason why the positive and negative preference scores in NaSPO cannot be combined into one metric? (Some additional intuition and clarification in the paper about this might be nice)
2.

**Limitations:**

Yes

**Strengths And Weaknesses:**

Soundness: While no rigorous theoretical proofs were made, claims are well justified through experimental results which show that PortraitRL performs well against benchmarks. For subjective matters like how much a generated image adhered to the prompt or kept detail, human users and evaluation metrics based on LVLM were used. Human users were not able to be used at scale but their judgements were noted to match the LVLM metrics. The necessity of steps of the PortraitRL algorithm are also justified through the ablation study.

Presentation: The submission is clear and well structured. The overall narrative is easy to follow, and the techniques involved are explained in a clear way.

Significance: The way reinforcement learning methods were used is likely to inspire future methods in PPT and reinforcement learning, both important topics. Human annotation also has a large cost and need for it is a key limitation in machine learning. it seems like NaSPO is a promising algorithm which can help reduce the annotation requirements and has a large applicability beyond the current context.

Originality: This works adapts and innovates on existing techniques to create a significant new algorithm to the PTT problem. NaSPO innovates and has improvements on DPSO.

---

> ### Author Rebuttal · Authors · 2026-03-31
>
> **Q1: Additional intuition and clarification about NaSPO.**
>
> **A1:** Thank you for the insightful question. In NaSPO, the preferred and non-preferred samples correspond to **two contrary optimization directions**: the optimization should move toward the preferred samples while moving away from the non-preferred ones (see Additional Derivation and Analysis as reference). To this end, we use two separate scores, namely positive and negative preference scores, to represent these two directions.:
>
> $$
> \begin{aligned}
> v^+ = v_{t+1} + \big(1 - \sigma(r(c, x_t) - Q_{\text{old}})\big)(v_{\theta,t+1} - v_{\text{old},t+1}), \\\\
> v^- = v_{t+1} + \big(1 + \sigma(Q_{\text{old}} - r(c, x_t))\big)(v_{\theta,t+1} - v_{\text{old},t+1}).
> \end{aligned}
> $$
>
> Specifically, in our method, for a generated image with advantage value $Adv$, we treat it as a positive sample with probability $Adv$ and as a negative sample with probability $1 - Adv$. Accordingly, the optimization direction for this generated image is computed as:
> $$
> Adv \cdot \lVert v_{\theta}-v^+\rVert_2^2 + (1 - Adv) \cdot \lVert v_{\theta}-v^-\rVert_2^2
> $$
> The effectiveness of this approach—combining positive and negative directions for optimization—has been validated in NFT.
>
> Overall, it is challenging to define two contrary directions using a single metric. Therefore, we adopt two separate scores to represent these two directions, and combine their optimization signals via the in-group advantage, resulting in a more stable optimization process.
>
> **Additional Derivation and Analysis:**
>
> As introduced in Section 3 of our paper, the positive preference score is derived from the score function:
> $$
> \nabla_{x_t} \log p_\theta (x_t | c, y_{\text{pref}}) = \nabla_{x_t} \log p (x_t | c) + \gamma \nabla_{x_t} \log p(y_{\text{pref}} | x_t, c).
> $$ The negative preference score is designed to guide the optimization toward directions that are equally consistent with human preferences, using the probability of non-preferred samples $p(y_{\text{non-pref}} | x_t, c)$. Starting from the fundamental identity $p(y_{\text{pref}}) = 1 - p(y_{\text{non-pref}})$, we can expand the gradient $\nabla_{x_t} \log p(y_{\text{pref}} | x_t,c)$ as follows:
> $$
> \begin{aligned}
> \nabla_{x_t} \log p(y_{\text{pref}} | x_t,c) &= \nabla_{x_t} \log(1 - p(y_{\text{non-pref}} | x_t, c)) = -\frac{\nabla_{x_t} p(y_{\text{non-pref}} | x_t,c)}{1 - p(y_{\text{non-pref}} | x_t,c)}.
> \end{aligned}
> $$ Since $\nabla_{x_t} \log p(y_{\text{non-pref}} | x_t,c) = \frac{\nabla_{x_t} p(y_{\text{non-pref}} | x_t,c)}{p(y_{\text{non-pref}} | x_t,c)}$, we can rewrite the gradient as:
> $$
> \nabla_{x_t} \log p(y_{\text{pref}} | x_t,c) = -w \nabla_{x_t} \log p(y_{\text{non-pref}} | x_t,c),
> $$ where the weighting coefficient $w = \frac{p(y_{\text{non-pref}} | x_t,c)}{1 - p(y_{\text{non-pref}} | x_t,c)}$. This formulation naturally leads to the negative preference score:
> $$
> \nabla_{x_t} \log p_\theta (x_t | c, y_{\text{pref}}) = \nabla_{x_t} \log p (x_t | c) - \gamma w \nabla_{x_t} \log p(y_{\text{non-pref}} | x_t, c).
> $$ Intuitively, the resulting negative sign explicitly models a repelling force—a "push" away from the non-preferred data manifold. From the derivation above, we can see that the positive and negative preference scores form a contrastive pair of preference directions.

---

> > ### Author Rebuttal · Reviewer_mvmP · 2026-04-04
> >
> > I thank reviewers for the thorough rebuttals. My questions are addressed and I maintain my original rating.

---

### Official Review · Reviewer_rNtA · 2026-03-12

**Soundness:** 2
**Presentation:** 2
**Significance:** 2
**Originality:** 2
**Overall Recommendation:** 3
**Confidence:** 5

**Summary:**

This paper proposes PortraitRL, a novel post-training framework that addresses these challenges with a multi-objective reward mechanism. Specifically, we employ LVLMbased reward functions to effectively evaluate PPT’s two challenges and apply within-group
standardization to eliminate scale differences, allowing these rewards to effectively guide optimization.

**Compliance With Llm Reviewing Policy:**

Affirmed.

**Final Justification:**

Compared with DiffusionNFT, I am not yet convinced that NaSPO represents a fundamentally stronger theoretical framework. While the authors argue that NFT can be viewed as a special case of NaSPO, the current presentation appears closer to a reparameterization or reinterpretation of the same positive/negative optimization paradigm rather than a rigorously established theoretical generalization. The appendix does provide additional derivation for the $\sigma(·)$ formulation, but I do not think it fully resolves the concern. **In my view, the appendix mainly explains how this form can be derived or rewritten under the authors’ modeling framework, rather than why it is inherently more reasonable, necessary, or superior to the manually tuned coefficient used in NFT.** In particular, the derivation still relies on several key assumptions that are not rigorously justified.

The paper essentially replaces a hyperparameter with $\sigma$, but it does not provide a systematic justification for why this particular sigmoid formulation should be considered optimal. The appendix offers only a non-rigorous argument rather than a strict proof.

**Key Questions For Authors:**

Please see the weaknesses.

**Limitations:**

This paper is likely a application of DiffusionNFT.

**Strengths And Weaknesses:**

Strengths:
1. This paper proposes PortraitRL, a novel post-training framework for PPT with a MultiObjective Reward Mechanism.
2. This paper also devises Negative-aware Score Preference Optimization (NaSPO), a novel GRPO-based RL algorithm optimized with both positive and negative preference scores.

Weaknesses:
1. The core method appears to be a direct application of the Diffusion-NFT pipeline to a Multi-Objective Reinforcement Learning setting.
2. It remains unclear whether the RL-optimized model can maintain Identity (ID) consistency under large-angle rotations.
3. The integration of an LVLM as a real-time evaluator in the RL loop introduces significant computational bottlenecks.

---

> ### Author Rebuttal · Authors · 2026-03-31
>
> **Q1: Difference between NaSPO and DiffusionNFT.**
>
> A1: Yes, our NaSPO does share some similarity with DiffusionNFT (shorted as NFT below), as both leverage positive and negative directions. However, our NaSPO is built upon a completely different theoretical foundation, and in fact, as we show below, **NFT can be viewed as a special case of NaSPO**.
>
> The key distinction lies in the underlying assumption. For a generated image with reward $Adv$, NFT interprets $Adv$ as the probability of the sample being a **"positive"** sample, while NaSPO interprets $Adv$ as the probability of the sample being a **"preferred"** sample, since our reward is based on an LVLM that simulates user preferences. This distinction is crucial: **the preference assumption enables us to naturally deploy the preference optimization framework -- DSPO** and derive reward-adaptive positive and negative directions, with a theoretical grounding that NFT's "positive sample" assumption lacks.
>
> Based on the above assumptions, NFT defines the positive and negative directions in an intuitive manner:
> $$
> \begin{aligned}
> v^+ = v_{\text{old},t+1} + \frac{1}{\beta}(v_{t+1} - v_{\text{old},t+1}), \\\\
> v^- = v_{\text{old},t+1} + \frac{1}{\beta}(v_{\text{old}t+1} - v_{t+1}).
> \end{aligned}\tag{1}
> $$ In NaSPO, we follow the formulation and theory of DSPO to define the positive and negative directions as positive and negative preference scores, which are given by:
> $$
> \begin{aligned}
> v^+ = v_{t+1} + \big(1 - \sigma(r(c, x_t) - Q_{\text{old}})\big)(v_{\theta,t+1} - v_{\text{old},t+1}), \\\\
> v^- = v_{t+1} + \big(1 + \sigma(Q_{\text{old}} - r(c, x_t))\big)(v_{\theta,t+1} - v_{\text{old},t+1}).
> \end{aligned}\tag{2}
> $$ From the above analysis, we can draw two key observations: **(1)** NaSPO is grounded in a more solid theoretical foundation, leading to a more principled definition of positive and negative directions. **(2)** The positive and negative directions in NaSPO do not rely on any hyperparameter — the direction magnitudes are dynamically modulated by the reward signal, resulting in more stable and efficient training. This is also empirically verified in Table 1 of our main paper.
>
> Moreover, as discussed in the Appendix, we can reformulate NaSPO into an NFT-like form:
> $$
> \begin{aligned}
> v^+ &= v_{\text{old},t+1} + \frac{1}{\sigma(r(x_t,c))}(v_{t+1} - v_{\text{old},t+1}), \\\\
> v^- &= v_{\text{old},t+1} - \frac{1}{\sigma(-r(x_t,c))}(v_{t+1} - v_{\text{old},t+1}),
> \end{aligned}\tag{3}
> $$ where
> $$
> r(x_t, c) = \beta_t \big( \lVert v_\theta(x_{t+1}, t+1) - v_{t+1} \rVert_2^2 - \lVert v_{\text{old}}(x_{t+1}, t+1) - v_{t+1} \rVert_2^2 \big).
> $$
>
> Comparing Eq. (1) and Eq. (3), we can see that NFT employs a fixed, hand-tuned hyperparameter $\beta$, whereas NaSPO adopts a sample-adaptive counterpart, leading to superior stability and efficiency. Notably, NFT can be viewed as a special case of NaSPO when the reward degenerates to a constant.
>
> To summarize, although NaSPO and NFT share a similar positive-negative direction framework, **NaSPO is built on a more rigorous theoretical foundation with reward-aware, sample-adaptive direction modulation**, resulting in more principled optimization, greater training stability, and consistently better performance.
>
> **Q2: Concerns about ID consistency under large-angle rotations.**
>
> **A2:** Good question! As demonstrated in Figure 1 (https://anonymous.4open.science/r/hatest-E02C/Figure1.png), even under large body-angle transformations or significant posture changes, the generated images accurately preserve the facial features, hairstyle, and overall appearance of the reference. For instance, in the first row, although the model alters gaze and body posture, the identity remains clearly recognizable. Similarly, in the second row, transitions from a frontal upright pose to a back-facing full-body shot, or from a prone to a frontal position, retain the key identity features while naturally adapting to the new pose. These results demonstrate that our RL optimization effectively maintains identity consistency across diverse rotations and posture variations.
>
> **Q3: Concerns about the  computational bottlenecks brought by LVLM.**
>
> **A3:** Thanks. Note that using an LVLM as a reward model is a standard practice in current methods, and any associated computational cost is a common limitation rather than specific to our work. Moreover, in our implementation, the LVLM only needs to output a single score in a single forward pass, without additional chain-of-thought or intermediate reasoning steps. As a result, the impact on training speed is acceptable, and the whole RL pipeline remains efficient.

---

> > ### Author Rebuttal · Reviewer_rNtA · 2026-04-03
> >
> > Thank you for your reply. Since the authors claim that NFT can be regarded as a special case of NaSPO, I am curious how the two methods compare on image generation benchmarks, such as GenEval and PickScore.

---

> > > ### Author Response · Authors · 2026-04-07
> > >
> > > We sincerely thank you for this insightful question. Note that we focus on developing effective RL methods (like NaSPO) for image editing in this work. The reported results have demonstrated the effectiveness of NaSPO in image editing. As suggested, to show the generalizability of NaSPO, we futher evaluate it on standard image generation benchmarks (GenEval and PickScore) in the table below. Due to the limited rebuttal time, we can only provide the results of NaSPO and NFT up to 80 training steps, with all the other parameters kept strictly consistent with those in the original NFT paper. It can be seen that our NaSPO still outperforms NFT in image generation tasks. This clearly shows the good generalizability of NaSPO.
> > >
> > > | Method | # Iterations| GenEval | PickScore |
> > > |---|:---:|:---:|:---:|
> > > | NFT | 80 | 0.88 | 22.8 |
> > > | NaSPO | 80 | **0.90** | **23.0** |

---

### Official Review · Reviewer_B5KN · 2026-03-13

**Soundness:** 2
**Presentation:** 2
**Significance:** 2
**Originality:** 2
**Overall Recommendation:** 4
**Confidence:** 3

**Summary:**

This paper proposes PortraitRL, a post-training reinforcement learning framework for Portrait Pose Transfer (PPT) to address two challenges of details preservation and prompt following. The authors introduce a Multi-Objective Reward Mechanism that uses Large Vision-Language Model (LVLM) logit-based scoring to evaluate both dimensions, combined with within-group Z-score standardization to balance the two reward signals and prevent reward hacking. They further propose Negative-aware Score Preference Optimization (NaSPO), a GRPO-based RL algorithm that automatically identifies positive and negative preference samples through within-group relative advantages, eliminating the need for annotated preference data. NaSPO defines both a positive preference score (steering toward desirable outputs) and a negative preference score (repelling from undesirable outputs via geometric reflection). Experiments on the CHEESE dataset show improvements over SFT and competing RL methods (FlowGRPO, DiffusionNFT) on LVLM-based evaluation metrics (Qwen-DP and Qwen-PF), with a user study of 5 participants on 50 samples confirming these trends.

**Compliance With Llm Reviewing Policy:**

Affirmed.

**Final Justification:**

all concerns have been addressed

**Key Questions For Authors:**

See weakness.

**Limitations:**

Yes

**Strengths And Weaknesses:**

Strengths：

1.The paper is well-written and easy to follow.

2.Comprehensive ablation studies validate each design choice and show the advantage of the proposed LVLM-based reward.

Weaknesses：

1.The motivation is not clearly separated from general image editing, and the method difference is still vague.

2.The evaluation is circular, since both training rewards and main metrics use the same LVLM family.

3.The evaluation scope is narrow, using only one dataset, one backbone, and one task.

4.Several key hyperparameters and training details are missing, which hurts reproducibility.

5.The baseline comparison is incomplete, with several recent editing and diffusion RL methods missing.

---

> ### Author Rebuttal · Authors · 2026-03-31
>
> **Q1: Concerns about motivation and method difference.**
>
> **A1:** Thank you for raising this concern. Compared to general image editing (e.g. ImgEdit), PPT poses two unique challenges: (a) **More complex editing requirements.** As shown in Figure 1 (https://anonymous.4open.science/r/seatest-E825/Figure1.png), existing image editing tasks typically involve simple, localized modifications, whereas PPT requires simultaneous changes across multiple aspects, including but not limited to pose, camera distance, and subject angle. (b) **Fine-grained detail preservation** under complex transformations. Beyond handling complex modifications, PPT further demands that identity-specific details such as facial features, makeup, and clothing remain consistent.
>
> To address these two challenges, we propose PortraitRL with two key designs: (a) We employ an LVLM to **simulate user preferences**, scoring both instruction-following ability and detail preservation as reward signals. (b) We propose **NaSPO**, which guides the optimization process by encouraging the model to approach positive preference scores while steering away from negative ones, leading to more stable training.
>
> **Q2: Concerns about circular evaluation with shared LVLM-based rewards and metrics.**
>
> **A2:** Thanks. In the table below, we add GPT-4o-based evaluation results. It shows consistent conclusions when GPT-4o is used instead of Qwen-VL. We will include the new metrics in the revision.
>
> | Method | Qwen-DR | Qwen-PF | GPT-DR | GPT-PF |
> |---|:---:|:---:|:---:|:---:|
> | Kontext+SFT | 0.8717 | 0.7928 | 0.8915 | 0.8515 |
> | +FlowGRPO| 0.8783 | 0.8138 | 0.8955 | 0.8625 |
> | +DiffusionNFT | 0.8894 | 0.8067 | 0.9040 | 0.8590 |
> | +PortraitRL | **0.8929** | **0.8260** | **0.9210** | **0.8800** |
>
> **Q3: Concerns about evaluation scope.**
>
> **A3:** Thanks for the suggestion. Beyond portrait photo transformation, PortraitRL also demonstrates strong performance on general editing tasks. In the table below, we report zero-shot comparisons between PortraitRL and baselines on the ImgEdit dataset. The results show that PortraitsRL still achieves the highest average score.
>
> | Method | Action | Adjust | Style | Background | Extract | Remove | Replace | Add | Compose | Avg |
> |:---|:---:|:---:|:---:|:---:|:---:|:---:|:---:|:---:|:---:|:---:|
> | FlowGRPO | 0.7833 | 0.8894 | 0.7740 | 0.9383 | **0.5265** | **0.6930** | 0.9500 | 0.9537 | **0.7043** | 0.7980|
> | DiffusionNFT | 0.8611 | 0.8723 | **0.8040** | 0.9426 | 0.4838 | 0.5744 | 0.9478 | 0.9411 | 0.6522 | 0.7772 |
> | PortraitRL | **0.9222** | **0.9000** | **0.8040** | **0.9766** | 0.5179 | 0.6116 | **0.9587** | **0.9663** | 0.6609 | **0.8074** |
>
> Furthermore, as shown in Figure 2 (https://anonymous.4open.science/r/seatest-E825/Figure2.png), PortraitRL exhibits excellent generalization to tasks beyond portrait editing, such as object editing (e.g., transform the building into a TV tower).
>
> Overall, we have validated the effectiveness of our PortraitRL on more datasets and more editing tasks. Due to the limited rebuttal time, more results with larger backbones can only be added in the next rebuttal stage.
>
> **Q4: Key hyperparameters and training details.**
>
> **A4:** Thanks for the thoughtful guidance. In the table below, we provide all the hyperparameters used in our paper.
>
> | Hyperparameters| Value |
> |:---|:---|
> | **Lora**: $r$ | 64 |
> | **Lora**: $\alpha$ | 128 |
> | **Basic**: Learning rate | 2e-4 |
> | **Basic**: $\beta_1$ | 0.9 |
> | **Basic**: $\beta_2$ | 0.999 |
> | **Basic**: Batch Size | 1 |
> | **Basic**: EMA Decay | 0.9 |
> | **Sampling**: Sampling Inference Steps | 25 |
> | **Sampling**: Resolution | 512x512 |
> | **Sampling**: Number of Images Per Prompt | 8 |
> | **Sampling**: Number of Groups | 48 |
> | **Reward**: $w_{dr}$ | 1.0 |
> | **Reward**: $w_{pf}$ | 1.0 |
>
> **Q5: Concerns about the incomplete baseline comparison.**
>
> **A5:** Thanks. In the table below, we present extra results using the latest editing method UniWorld-V2. We find that our PortraitRL outperforms Uniworld-V2 in terms of the average of Qwen-DR and Qwen-PF. Due to the limited rebuttal time, more baselines can only be included in the next rebuttal stage.
>
> | Method | CLIP-I | DINO-I | CLIP-T | Qwen-DR | Qwen-PF |
> |--------|--------|--------|--------|---------|---------|
> | Kontext | 0.8482 | 0.8012 | **0.4417** | 0.7824 | 0.7529 |
> | +SFT | 0.8721 | 0.8163 | 0.4363 | 0.8717 | 0.7928 |
> | &nbsp;&nbsp;+FlowGRPO | 0.8720 | 0.8213 | 0.4364 | 0.8783 | 0.8138 |
> | &nbsp;&nbsp;+DiffusionNFT | **0.8768** | **0.8268** | 0.4400 | 0.8894 | 0.8067 |
> | &nbsp;&nbsp;+Uniworld-V2 | 0.8693 | 0.8174 | 0.4396 | 0.8656 | **0.8271** |
> | &nbsp;&nbsp;**+PortraitRL** | 0.8757 | 0.8239 | **0.4417** | **0.8929** | 0.8260 |

---

> > ### Author Rebuttal · Reviewer_B5KN · 2026-04-03
> >
> > all concerns have been addressed

---

### Official Review · Reviewer_MRgv · 2026-03-13

**Soundness:** 3
**Presentation:** 3
**Significance:** 3
**Originality:** 3
**Overall Recommendation:** 4
**Confidence:** 3

**Summary:**

This paper proposes PortraitRL, a multi-objective reward reinforcement learning framework for portrait pose transfer. A multi-objective reward mechanism is proposed, with a LVLM-based score function used to ensure fine-grained evaluation, and a within-group standardization strategy presented to ensure the effectiveness of both prompt following and detail preservation objectives. Additionally, a NaSPO reinforcement learning algorithm is tailored for optimization without the need for annotated preference data. Extensive experiments demonstrate the effectiveness of the proposed method.

**Compliance With Llm Reviewing Policy:**

Affirmed.

**Final Justification:**

My main concerns have been adequately addressed, so I am keeping my original rating.

**Key Questions For Authors:**

Please see the weaknesses.

**Limitations:**

The authors are encouraged to include discussions on the limitations (e.g., extreme input poses or illumination conditions) and potential negative societal impact of the proposed method.

**Strengths And Weaknesses:**

## Strengths
1. The paper is clearly organized and well-presented, which is easy-to-follow.
2. The proposed multi-objective reward mechanism and the NaSPO algorithm are well-motivated. The authors also include extensive experiments to demonstrate the efficacy of the method.
3. The visualization results are impressive.

## Weaknesses
1. Although the proposed method performs well on the CHEESE dataset, to ensure a more comprehensive evaluation on the method's effectiveness, the authors should test it on a wider variety of other real-world data with various lighting, backgrounds, and human appearances. This is crucial for assessing the method's generalization ability under different scenarios.
2. Did the authors notice any failure cases of the method? The corresponding limitations should be discussed.
3. The framework involves some key hyper-parameters, such as the reward weights w_dr and w_pf, but the corresponding sensitivity analysis was not performed. The potential impact of fluctuations in these parameters on the method's performance should be discussed.
4. As a post-training framework, its training cost is one of the important factors in practical applications. The authors should provide a discussion of the framework's computational complexity.

---

> ### Author Rebuttal · Authors · 2026-03-31
>
> **Q1: Concerns about generalization ability.**
>
> **A1:** Thanks for the valuable suggestion! Our explanantions are three-fold:
>
> (1) In the table below, we additionally report zero-shot comparisons between PortraitRL and other baselines on ImgEdit, another real-world general editing dataset. It can be seen that PortraitRL achieves the best average performance, further demonstrating the effectiveness of our proposed method.
>
> | Method | Action | Adjust | Style | Background | Extract | Remove | Replace | Add | Compose | Avg |
> |:---|:---:|:---:|:---:|:---:|:---:|:---:|:---:|:---:|:---:|:---:|
> | FlowGRPO | 0.7833 | 0.8894 | 0.7740 | 0.9383 | **0.5265** | **0.6930** | 0.9500 | 0.9537 | **0.7043** | 0.7980|
> | DiffusionNFT | 0.8611 | 0.8723 | **0.8040** | 0.9426 | 0.4838 | 0.5744 | 0.9478 | 0.9411 | 0.6522 | 0.7772 |
> | PortraitRL | **0.9222** | **0.9000** | **0.8040** | **0.9766** | 0.5179 | 0.6116 | **0.9587** | **0.9663** | 0.6609 | **0.8074** |
>
> (2) In Figure 1 (https://anonymous.4open.science/r/anotest-FE13/Figure1.png), we present visualization examples from the action task of ImgEdit. We can observe that: (a) PortraitRL performs well across diverse backgrounds (solid color, natural, stage), human appearances (gender, age, hair color), and lighting conditions (natural light and stage light). (b) Beyond the simple instructions in the original data, we additionally design more complex instructions. We find that PortraitRL is able to understand and follow complex editing requirements, better aligning with real-world user editing needs.
>
> (3) We further conduct experiments on non-portrait editing scenarios. As shown in Figure 2 (https://anonymous.4open.science/r/anotest-FE13/Figure2.png), although PortraitRL is trained on portrait photo data, it still demonstrates excellent instruction-following ability and detail preservation when the image contains no human portraits and only involves object-level modifications (e.g., "Remove the pearl necklace"). As suggested, we will include these analyses and results.
>
> **Q2: Failure cases and limitations.**
>
> **A2:** Thank you for the helpful guidance! Indeed, we observe that PortraitRL sometimes suffers from artifacts when the editing task is highly challenging. From Figure 3 (https://anonymous.4open.science/r/anotest-FE13/Figure3.png), we can see that: (a) When the instruction requires expanding the human body (e.g., transforming a close-up portrait into a full-body view), PortraitRL occasionally produces results with distorted body proportions. (b) When the instruction involves difficult pose changes, such as requiring a girl to transition from sitting to standing behind a boy, PortraitRL sometimes takes a shortcut by adding a duplicate of the same person instead of performing the intended modification. To address these artifacts, future work will explore introducing a third reward component, such as an aesthetics-aware or artifact-penalizing reward, to better preserve structural integrity. We will incorporate this analysis into the revision.
>
> **Q3: Concerns about key hyperparameters.**
>
> **A3:** Thanks. In the table below, we present results trained with different hyperparameters $w_{dr}$ and $w_{pf}$. We can see that increasing the weight of either $w_{dr}$ or $w_{pf}$ leads to a moderate decrease in the score of the other metric. As suggested, we will include these analyses and results.
>
> | w_dr : w_pf | CLIP-I | DINO-I | CLIP-T | Qwen-DR | Qwen-PF | Qwen-AVG |
> |:---:|:---:|:---:|:---:|:---:|:---:|:---:|
> | 1.0:1.5 | 0.8747 | 0.8251 | 0.4420 | 0.8897 | 0.8269 | 0.8583 |
> | 1.5:1.0 | 0.8807 | 0.8300 | 0.4389 | 0.8958 | 0.7983 | 0.8471 |
> | 1.0:1.0 | 0.8760 | 0.8250 | 0.4417 | 0.8953 | 0.8280 | **0.8617** |
>
> **Q4: Concerns about training cost.**
>
> **A4:** Good question! We provide a detailed comparison of computational complexity in the table below. All three methods share the same trainable parameter ratio (1.4414\%), indicating that PortraitRL introduces no additional trainable parameters. Moreover, PortraitRL exhibits nearly identical peak memory, sampling time, optimization time, and throughput compared to DiffusionNFT, demonstrating that our NaSPO introduces negligible computational overhead. We will add this result in the revision.
>
> | Method | Trainable Param Ratio  | Peak Mem | Sample Time | Optim Time | Throughput |
> |:---|:---:|:---:|:---:|:---:|:---:|
> | FlowGRPO | 1.44% | 47.61 GB | 168.85s | 379.23s | 0.70 |
> | DiffusionNFT | 1.44% | 47.95 GB | 190.30s | 665.46s | 0.45 |
> | PortraitRL | 1.44% | 47.95 GB | 190.82s | 662.98s | 0.45 |

---

> > ### Author Rebuttal · Reviewer_MRgv · 2026-04-03
> >
> > I appreciate the detailed responses from the authors. Most of my concerns are addressed and I keep my original rating.

---

### Decision · Program_Chairs · 2026-04-30

**Decision:**

Accept (regular)

**Comment:**

This paper proposes PortraitRL, a post-training reinforcement learning framework for portrait pose transfer (PPT) that introduces (1) a multi-objective reward mechanism using LVLM-based scoring with within-group Z-score standardization and (2) Negative-aware Score Preference Optimization (NaSPO), a GRPO-based RL algorithm that leverages both positive and negative learning signals without requiring annotated preference data.

It received scores of 5/4/4/3 from four reviewers. All reviewers acknowledged that the paper is well-written and clearly presented, and the most critical reviewer (rNtA, score 3) argued that NaSPO is essentially a direct application of DiffusionNFT and questioned ID consistency under large rotations and LVLM computational overhead.

Overall, the authors' contribution to be meaningful in aggregate: the reward-adaptive, hyperparameter-free direction modulation derived from the DSPO preference framework is a non-trivial design choice that is empirically validated across multiple benchmarks. The multi-objective reward design with within-group standardization is well-motivated for the PPT setting. The rebuttal successfully addressed concerns about evaluation scope, circular metrics, baselines, and reproducibility. The remaining weakness is that the theoretical novelty over DiffusionNFT is incremental rather than fundamental, and the paper would benefit from a more honest positioning of this relationship.

Therefore, the AC is happy to accept this paper to ICML. Congratulations! Please keep in mind that the authors are strongly encouraged to incorporate the reviewers' feedback to prepare for a solid camera-ready version.